# Talk in Pieces, See in Whole: Disentangling and Hierarchical Aggregating Text Representations for Language-based Object Detection

## Abstract

While vision-language models (VLMs) have made significant progress in multimodal perception (e.g., open-vocabulary object detection) with simple language queries, state-of-the-art VLMs still show limited ability to perceive complex queries involving descriptive attributes and relational clauses. Our in-depth analysis shows that these limitations mainly stem from text encoders in VLMs. Such text encoders behave like bags-of-words and fail to separate target objects from their descriptive attributes and relations in complex queries, resulting in frequent false positives. To address this, we propose restructuring linguistic representations according to the hierarchical relations within sentences for language-based object detection. A key insight is the necessity of disentangling textual tokens into core components—objects, attributes, and relations ("talk in pieces")—and subsequently aggregating them into hierarchically structured sentence-level representations ("see in whole"). Building on this principle, we introduce the `TaSe` framework with three main contributions: (1) *a hierarchical synthetic captioning dataset* spanning three tiers from category names to descriptive sentences; (2) *Talk in Pieces*, the three-component disentanglement module guided by a novel disentanglement loss function, transforms text embeddings into subspace compositions; and (3) *See in Whole*, which learns to aggregate disentangled components into hierarchically structured embeddings with the guide of proposed hierarchical objectives. The proposed `TaSe` framework strengthens the inductive bias of hierarchical linguistic structures, resulting in fine-grained multimodal representations for language-based object detection. Experimental results under the OmniLabel benchmark show a 24% performance improvement, demonstrating the importance of linguistic compositionality.

## 1 Introduction

Vision-language (VL) understanding, which aims to perceive each modality and form associations between them, is a long-standing and fundamental problem. Recently, foundational VLMs such as CLIP (Radford et al., 2021a) have leveraged web-scale image-text pairs to learn generic VL representations, achieving strong generalization performance on tasks like image classification and image-text retrieval. Building upon these advances, recent studies have actively explored grounding language queries into specific image regions (e.g., open-vocabulary object detection (Liu et al., 2024b; Zhao et al., 2024; Yin et al., 2025)). Many approaches (Liu et al., 2024a; Li et al., 2022b) distill the general VL knowledge embedded in foundational models into object detectors and have demonstrated remarkable results in detecting previously unseen object categories—commonly referred to as open-vocabulary object detection (Gu et al., 2021).

Despite these advances, current VL detectors often succeed only when the input queries are short and consist of simple category names. They still struggle to fully comprehend complex language queries and accurately localize the corresponding objects. To illustrate this limitation, we conduct a preliminary analysis using the state-of-the-art foundation model for visual grounding, GLEE (Wu et al., 2024) (see Fig. 1a). The model reliably detects objects given simple noun phrases (e.g., "segway"). However, it fails when faced with more complex and specific queries (e.g., "segway with a man"), indicating its limited compositional understanding.

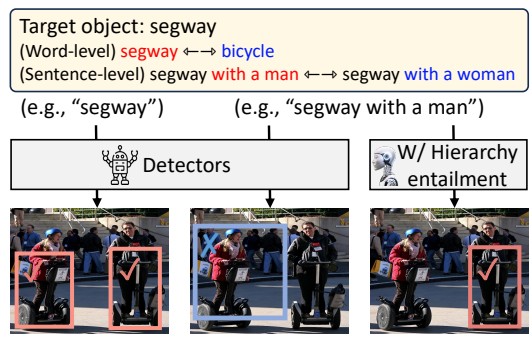

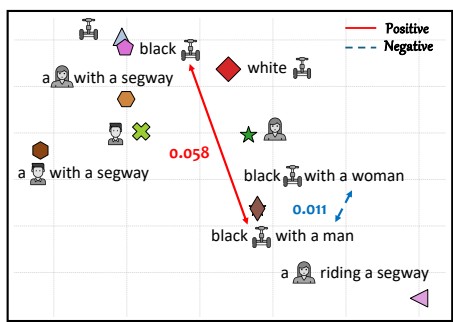

(a) Detection results from w/o and w/ our hierarchy entailment

(b) t-SNE of sentence features from text encoder of VLMs

Figure 1: (a) VL detectors struggle with attributes or relations due to limitations in capturing fine-grained semantics from image-text similarity. We propose a hierarchical multimodal representation learning to enhance the linguistic compositionality of complex language queries. (b) Limitations of text encoders in VLMs for compositional understanding. Although some sentences refer to different target objects, their embeddings remain close due to shared tokens, contrary to the ideal case in which they should be well-separated (e.g., "with a man" vs. "with a woman").

To investigate the underlying cause of this limitation, we visualize sentence-level text features using t-SNE (Van der Maaten & Hinton, 2008) in Fig. 1b. Interestingly, we observe that although some sentences ("a segway with a man" vs. "a segway with a woman") refer to different target objects ("with a man" vs. "with a woman"), their embeddings remain close in the feature space due to shared tokens ("segway"). The contrasts with the ideal cases, where embeddings of distinct target objects should be well-separated, while those of the same object with different attributes should be closer for object detection ("a segway" vs. "a black segway"). These results indicate limited hierarchical and compositional understanding in current VLMs (Yuksekgonul et al., 2022). Most VL detectors (Liu et al., 2024a; Li et al., 2022b) are trained to align a few positives with image regions while distinguishing them from negatives using contrastive learning. For example, training with augmented captions (Li et al., 2023b; Yuksekgonul et al., 2022) labeled as positive or negative is effective for coarse-grained concept analysis. Still, detectors often struggle to handle tasks that require understanding of fine-grained text, such as reasoning over interactions between objects (e.g., "bigger than"). Sentence embeddings obtained via simple pooling compress token-level information and fail to capture contextualization in language queries. Beyond augmentation, sentence embeddings need to disentangle text tokens and encode compositional features. We argue that VL detectors should go further: representations need to see the whole sentence from meaningful pieces.

In this paper, we propose a novel framework that disentangles component-wise text features ("Talk in Pieces") and explicitly learns hierarchical knowledge ("See in Whole") from these disentangled representations to construct sentence-level understanding for language-based object detection. We refer to our framework as **TaSe** (**Ta**lk in Pieces, **Se**e in Whole). We begin by formally defining a hierarchical structure tailored for language-based object detection. Specifically, we design our new HiVG dataset, a three-tiered hierarchy, object–attribute–relation, where the first tier represents object category names (i.e., segway), the second tier adds descriptive attributes (i.e., black), and the last tier includes relational phrases that describe interactions or contexts (i.e., with a woman). We also construct negative samples following the same three-tier structure. Tier 1 leverages Word-Net (Miller, 1995) and ImageNet (Deng et al., 2009) categories to sample non-target classes as negative samples. Tiers 2 and 3 employ BERT (Koroteev, 2021)-based masking to generate negative prompt variants by perturbing parts of the original text. This mitigates bag-of-words behavior driven by lexical overlap and supports reliable semantic discrimination between positive and negative prompts. Our approach builds on phrase grounding datasets like Visual Genome (Krishna et al., 2017), which provide densely annotated phrases associated with images and object regions. Using a large language model (LLM) (Dubey et al., 2024; Yang et al., 2025; OpenAI, 2025), we abstract these phrases into a three-tier hierarchy—object, attribute, relation—by sequentially removing relational and attribute information in phrases to obtain the final object categories. Unlike typical generation-based approaches (e.g., generating sentences from category names (Li et al., 2023b) or captions from images), our abstraction-based process performs reverse abstraction, effectively mitigating hallucination issues (Ji et al., 2023) common in generative models.

To effectively construct contextualized (i.e., holistic) sentence representations from the HiVG dataset, we disentangle into several key aspects ("Talk in Pieces")—such as objects, attributes, and relations. This design allows us to disentangle text representations into subspaces to adjust targeted token embeddings and preserve meaningful information in the remaining features. For this purpose, we further design a lightweight learnable attention module for the TriDe (Three-component disentanglement), enabling efficient fine-tuning of conventional text encoders. The key idea of TriDe is to leverage the hierarchical structure of the HiVG dataset to contrast component-wise tokens so that targeted tokens to be adjusted without loss of meaningful information.

Then, we guide the model to learn linguistic representations that capture these levels of abstraction. This facilitates learning of sentence context enriched with descriptive attributes and relational clauses. We introduce a hierarchical aggregation method ("See in Whole") based on sentence-level hierarchy entailment, which effectively models sentence-level hierarchical relationships with our HiVG. Our learning hierarchical objective offers a richer and more structured alternative to naïve contrastive learning, which typically aligns image regions with positive tokens in a sentence while contrasting them with negative tokens. In contrast, our method models the full sentence hierarchy, promoting a more dense VL understanding.

To summarize, our main contributions are as follows: 1) We present an efficient hierarchical data generation pipeline that abstracts dense existing phrases into an explicit hierarchical structure of "object–attribute–relation." 2) We introduce a novel framework for disentangling the three core components and employ the TriDe loss to guide this process. 3) We propose a method for learning disentangled and hierarchical representations that capture sentence-level inductive biases and can be integrated into conventional VL detectors. With hierarchical learning on our generated dataset HiVG, our model significantly outperforms strong baselines, including state-of-the-art VL detectors, on challenging language-based object detection benchmarks such as OmniLabel (Schulter et al., 2023) and D3 (Xie et al., 2023).

## 2 RELATED WORKS

### 2.1 LANGUAGE-BASED OBJECT DETECTION

Language-based object detection aims to locate and identify objects in images using free-form text. One of the leading approaches is to transfer the pre-trained model and align images and texts using contrastive learning (Gao et al., 2024; Li et al., 2022b; 2023b; Park et al., 2024). Contrastive learning enhances compositionality in VLMs by capturing relationships with contextual entities and improves the understanding of object relationships (Li et al., 2023b; Minderer et al., 2022a; Gu et al., 2021; Liu et al., 2024b). GLIP (Li et al., 2022b) proposes to add deep fusion layers between different modalities and learn a language-aware visual representation based on reformulated alignment scores.

However, existing approaches overlook the need for contextualized sentence-level understanding of VL text embedding. For example, APE (Shen et al., 2024); GLEE (Wu et al., 2024); Dino-x (Ren et al., 2024); and Zeng et al. (2024) explore VLM alignment challenges and highlight the need to improve reasoning capabilities in multimodal LLMs. These works investigate model capabilities from restricted VL perspectives, with a primary focus on fine-grained textual details and inter-object relationships. VL detectors still struggle to align images with syntactically intricate language queries (Wang et al., 2023), underscoring the need for a more grounded contextual understanding of text.

Disentangled representation learning is a method for enhancing linguistic understanding by learning fine-grained representations (Bengio et al., 2013; Wang et al., 2024). Several approaches have been proposed for disentangled representation learning, including prompt learning (Lu et al., 2023; Zheng et al., 2024), learnable vectors (Qi et al., 2024), and attention-based mechanisms (Wu et al., 2025). Prior works have introduced methods for designing object–attribute compositions, which improve compositional zero-shot learning. In contrast to these concept-aware approaches that disentangle objects and attributes for recomposition, our method leverages disentanglement to capture hierarchical sentence structures and contextualized understanding.

### 2.2 HIERARCHICAL ENTAILMENT FOR VISION-LANGUAGE MODELS

To better represent the embedding space of VLMs, hyperbolic learning has highlighted the need to capture hierarchical structures and relationships in multimodal data. Hyperbolic learning was

formulated on the Poincaré ball by Ganea et al. (2018), learning entailment relations between embedded objects. The formulation now extends the more common Lorentz model as Lou et al. (2020) due to its computationally heavy Gyrovector operations. Hyperbolic learning maps the embedding into an entailment cone (EC) to represent hierarchical entailment in a continuous space. Recent studies investigated the use of the EC embedding for vision tasks (Atigh et al., 2022; Kong et al., 2024; Khrulkov et al., 2020), multimodal learning Desai et al. (2023); Hong et al. (2023); Pal et al. (2024), and synthetic data generation (Kong et al., 2024).

However, the hyperbolic manifold needs to transpose features from Euclidean to hyperbolic and requires additional hyperparameter configurations. To address this limitation, Alper & Averbuch-Elor (2024) proposed radial embedding (RE) optimization for learning hierarchical representations directly in Euclidean space. Inspired by this approach, we extend RE optimization to language-based object detection based on hierarchical representation learning at the sentence level. While previous works explore hierarchical manifolds to capture natural hierarchy (Lang et al., 2022), sentence-level hierarchy objectives remain underexplored. This work introduces a hierarchical modeling approach to define the sentence-level hierarchy entailment with compositional learning, which captures inclusive relationships between hierarchy nodes in language-based object detection.

## 3 TaSe: Disentangled and Hierarchical Text Representation Learning for Language-based Object Detection

This section introduces `TaSe`, a framework for disentangling and hierarchy aggregating method. Specifically, our approach comprises three components: 1) the HiVG dataset (Sec. 3.1), a synthetic dataset re-captioned from VG to capture hierarchical entailment relations; 2) disentangling text representations into objects, attributes, and relations for a component-wise subspace for aligning semantic pieces within sentences (Sec. 3.2); and 3) a hierarchical aggregation method to represent contextualized sentence embedding based on disentangled tokens (Sec. 3.3). Fig. 2 outlines the `TaSe` to learn contextualized sentence representations within language-based object detection.

### 3.1 HiVG: Hierarchy Captioning Pipeline

Although augmented captions enhance fine-grained textual representations (Li et al., 2023b; Yuksekgonul et al., 2022), open-vocabulary detectors often rely on keywords and fail to separate target objects from their attributes and relations, owing to the absence of hard textual negatives that reflect linguistic hierarchy. To address this problem, we propose a **Hi**erarchical captioning pipeline that re-captions the **V**isual **G**enome dataset (**HiVG**) by leveraging LLMs (i.e., Llama-3-8B (Dubey et al., 2024), Qwen3-8B (Yang et al., 2025), and GPT5-o-nano (OpenAI, 2025)) and lexical databases (e.g., WordNet (Miller, 1995) and ConceptNet (Speer et al., 2017)). HiVG is a synthetic dataset constructed by spanning from category names to descriptive sentences and structuring hierarchical captions into three tiers: objects, attributes, and relations. Each caption in the Visual Genome (Krishna et al., 2017) annotation is transformed into three positive ($e^+$) and negative tiers ($e^-$) where $e$ follows the notation introduced in Sec. 3.3. We show an example for the input image in Fig. 2.

- Tier 1. Category names (object): containing the class name (e.g., **woman** ($e_1^+$) and **man** ($e_1^-$)).
- Tier 2. Enriched descriptions (w/ attribute): adding an attribute to the object (e.g., **middle** woman ($e_2^+$) and **left** woman ($e_2^-$) for learning fine-grained linguistic compositionality).
- Tier 3. Contextual understanding (w/ attribute and relation): emphasizing the relationships between objects by injecting a relation into the second-tier caption (e.g., middle woman **with dark hair** ($e_3^+$) and **with red shirt** ($e_3^-$)).

Further details of our re-captioning approach and examples are provided in the supplementary material (see Sec. A, Fig. 13).

### 3.2 Talk in Piece: Component-wise Text Disentanglement

Textual descriptions typically contain not only descriptive attributes but also complex relational structures, which cause false positives in language-based object detection. To address this, we propose the TriDe module to disentangle text embeddings into meaningful subspaces, which adaptively refines these components to enhance semantic representation.

**Text embedding.** We extract text features by CLIP text encoder with low-rank adaptation (LoRA) (Hu et al., 2021) for efficiently evolving text embedding from the text encoder. Let $\{v_i, t_i\}_{i=1}^{B}$ be a

Figure 2: The overall framework of TaSe. (Left) The text encoder is fine-tuned with LoRA (Hu et al., 2021) and the TriDe module to restructure text representations. (Right) **Top:** Hierarchy aggregated embeddings with HiVG, where the recaptioned dataset passes through the TriDe module to learn linguistic hierarchy. **Bottom:** Architecture of the TriDe module.

batch of image–text pairs. The text embedding $\mathbf{X} = \mathcal{T}_\theta(t_i)$, where $X \in \mathbb{R}^{B \times T \times d_{\text{model}}}$, is obtained by the text encoder of CLIP. Here, $B$, $T$, and $d_{model}$ denote the batch size, number of tokens, and embedding dimension, respectively. A text projection layer maps the input into $X \in \mathbb{R}^{B \times T \times D}$, where $D$ denotes the text embedding dimension.

**Component-wise disentanglement.** We disentangle text representations into three components—objects, attributes, and relations. This design mirrors the three-tier structure of HiVG and facilitates the learning of effective contextualized sentence embeddings. We adjust learnable vectors $V_O, V_A, V_R \in \mathbb{R}^{B \times T \times D}$ to disentangle the text embedding into the three components. We employ a multi-head cross-attention layer between the learnable vectors and text embedding X. Let FFN, LN, and Proj denote the feedforward network, layer normalization, and projection layer, respectively. The TriDe module is defined as follows:

$$\mathbf{X} = \text{LN}(\text{Proj}(X + \text{FFN}(X))),$$
$$[O, A, R] = \text{CrossAttn}(X, [V_O, V_A, V_R]), \quad (1)$$
$$\mathbf{E} = \text{pooling}(\text{FFN}(\text{LN}(O + A + R))),$$

where O, A, R, and E represent the object, attribute, relation components, and restructured text embedding. Note

Figure 3: Learning process for hierarchically structure. The model is trained with contrastive learning on HiVG, where text features are disentangled into subspaces across tiers and optimized with cosine distance.

that E is employed to learn hierarchical entailment for contextualized sentence representations in Sec. 3.3. The CrossAttn(Q, V) with the scaling factor $d_k$ is defined as follows:

$$\text{CrossAttn}(Q, V) = \text{Softmax}\left(\frac{\text{Proj}(Q)\text{Proj}(V)^T}{\sqrt{d_k}}\right)\text{Proj}(V). \quad (2)$$

Here, Q denotes the *query*. The *key* and *value* are identical in this formulation, so only the V term is used. The E is mean-pooled to encode the sentence-level representation used for alignment with visual embeddings. The detailed algorithm is described in Supplementary Alg. 1.

**Objective for component disentanglement.** We found that directly increasing the angular distance for negative captions amplifies the effect of shared lexical content, leading to unstable gradients and sharp loss escalation. This disentangle–aggregate design separates semantic components and recombines them, and yields stable optimization of angular relationships in the embedding space. In Fig. 3, components are aligned with their positive counterparts, while negatives are enforced to remain distant according to their tier. Let $t$ be the tier of HiVG, and the disentanglement of text embedding is adjusted $\mathcal{L}_{\text{TriDe}}$ as follows:

$$\mathcal{L}_{\text{TriDe}} = \lambda \underbrace{\sum_{(i,j) \in \{(O,A),(O,R),(A,R)\}} |\mathbf{i} \cdot \mathbf{j}|}_{\text{O–A–R Orthogonality}} + \underbrace{\sum_{t=1}^{l} \left(m + \cos(O_t^+, O_{t+1}^+) - \cos(O_t^+, O_t^-)\right)}_{\text{Object Disentanglement}}$$

$$+ \underbrace{\sum_{t=2}^{l} \left(m + \cos(A_t^+, A_{t+1}^+) - \cos(A_t^+, A_t^-)\right)}_{\text{Attribute Disentanglement}} + \underbrace{\left(m - \cos(R_l^+, R_l^-)\right)}_{\text{Relation Disentanglement}} \quad (3)$$

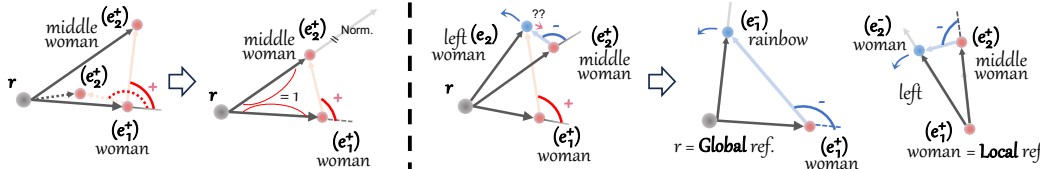

Figure 4: Limitation and solution of the RE objective. (left) Positive hierarchy objective. Solid - ideal angle, Dotted - distortion from larger upper-tier embedding. Angular distance defines sentence-level hierarchy by capturing directional differences independent of magnitude. (right) Negative hierarchy objective: Red = positive, Blue = negative. With the root fixed, the $e_i^-$ (i.e., "left woman") indicates a trade-off; when information is shared (i.e., woman), angles are measured relative to a negative-supportive embedding.

$\mathcal{L}_{\text{TriDe}}$ and $m$ represent compositional loss for the TriDe module and margin, respectively. $\lambda$ represents a hyperparameter for stability adjustment. Minimizing correlation among components promotes an inductive bias toward semantically grounded object representations.

### 3.3 SEE IN WHOLE: SENTENCE-LEVEL HIERARCHICAL AGGREGATION

We design a hierarchy aggregation method of disentangled features to serve fine-grained semantic distinction, which helps capture contextual meaning beyond simple word-level perception.

**Background: hierarchical entailment in Euclidean space.** The goal of hierarchical entailment is to learn general concepts by representing entailment relations via low-dimensional embeddings (Ganea et al., 2018). While conventional contrastive learning to learn embedding from pair-wise (i.e., positive and negative), the RE objective (Alper & Averbuch-Elor, 2024) aims to represent the hierarchy structure of embedding by exterior angle with respect to the reference point. Two key advantages of this representation learning with the RE are: 1) modeling sentence-level structure; and 2) learning compositional generalization without requiring transformation into sphere space. Let $\Xi$ denote the exterior angle in radians and $r \in \mathbb{R}^d$ denote a root embedding, the exterior angle between embedding $a$ and $b$ in the RE objective is defined:

$$\Xi\langle a, b\rangle = \cos^{-1}\left(\frac{\mathbf{a}' \cdot \mathbf{b}'}{\|\mathbf{a}'\|\|\mathbf{b}'\|}\right) \leq \pi.$$
$$\text{where} \quad \mathbf{a}' = \mathbf{a} - r, \quad \mathbf{b}' = \mathbf{b} - r - \mathbf{a}', \quad \mathbf{a}, \mathbf{b} \in \mathbb{R}^d \tag{4}$$

The value $\langle a, b\rangle_\Xi \in [0, \pi]$ is bounded. Given a root embedding $r$ and a reference embedding of a text embedding $e$, the objective function of the RE is represented as follows:

$$\mathcal{L}_{RE} = \sum \left(\langle e_i^+, e_{>i}^+\rangle_\Xi - \langle e_i^+, e_i^-\rangle_\Xi\right). \tag{5}$$

Here, $e_i^+$ and $e_{>i}^+$ denote distinct positive embeddings, and $e_i^-$ is a corresponding negative. The objective $\mathcal{L}_{RE}$ encourages smaller exterior angles between positive pairs while enforcing larger angles between positive and negative pairs. This deviation reflects a misalignment from the reference anchor $r$ (frozen) and corresponds to a larger angular distance in the embedding space.

**Reference-based hierarchy induction.** The previous approach still faces challenges in enhancing compositional generalization from the perspective of sentence-level hierarchy entailment (see Fig. 4). First, we introduce a regularization term to preserve the inherent embedding space and define two objective functions for hierarchy entailment. Since angular-only supervision may induce directional bias and degrade representational fidelity, we regularize the embedding space to retain knowledge. Second, contrastive learning is formulated using exterior angles, where sentence pairs are trained with respect to a reference-conditioned point. A key consideration in sentence-level contrastive learning with a text encoder is that a negative $e_t^-$ is also positive with respect to the previous tier $e_{t-1}^+$. To account for this, we position samples in opposing directions based on a dynamic reference $r$ rather than a fixed root (see Eq. 6).

**Hierarchical objectives.** Let $l$ denote the tier of HiVG in the object-attribute-relation hierarchy. Let $e_t$ be an embedding at tier $t$ of HiVG, $e_t^+$ its positive counterpart at tier $l$, and $e_t^-$ a negative sample at the same tier. We use two complementary losses—(i) alignment across tiers and (ii) within-tier

discrimination—defined compactly as

$$\mathcal{L}_{\mathrm{H}^+} = \underbrace{\sum_{t=1}^{l} \Xi \langle e_t^+, e_{t+1}^+ \rangle + \Xi \langle e_t^+, e_{t+1}^- \rangle}_{\text{Positive hierarchy}}, \quad b' = (b-r) - a',$$

$$\mathcal{L}_{\mathrm{H}^-} = \underbrace{\sum_{t=1}^{l} \Xi \langle e_t^+, e_t^- \rangle}_{\text{Negative hierarchy}}, \quad \underbrace{b' = (r-b) - a'}_{\text{Ref. update with neg.}}, \quad r = \underbrace{\begin{cases} r & t=0 \\ e_{t-1}^+ & t>0 \end{cases}}_{\text{Global / Local ref.}} \quad (6)$$

with normalization $e = \frac{e-r}{\|e-r\|+\epsilon}$, $e \in \{e^+, e^-\}$. Note that normalization maintains directional consistency independent of embedding scale. $r$ is adapted to expose directional differences not captured by global alignment and to account for locally meaningful variation. The proposed formulation minimizes directional deviation from the reference point, which helps preserve the intrinsic structure of the embedding space while enforcing separation between negative pairs.

**Training objectives.** For learning contextualized features, the objective function of hierarchical and disentangled representation learning, $\mathcal{L}_{\mathrm{TaSe}}$, is defined as:

$$\mathcal{L}_{\mathrm{TaSe}} = \mathcal{L}_{\mathrm{TriDe}} + \mathcal{L}_{\mathrm{H}^+} + \mathcal{L}_{\mathrm{H}^-}. \quad (7)$$

The loss function is formulated as a weighted sum of classification, localization, and hierarchy losses updated on the text encoder and VL fusion layers as follows:

$$\mathcal{L} = \mathcal{L}_{class} + \mathcal{L}_{bbox} + \mathcal{L}_{giou} + \mathcal{L}_{\mathrm{TaSe}}, \quad (8)$$

where $\mathcal{L}_{class}$ represents Focal loss (Lin et al., 2017), $\mathcal{L}_{bbox}$ represents L1 loss, and $\mathcal{L}_{giou}$ represents generalized intersection over union (GIoU) loss (Rezatofighi et al., 2019).

## 4 EXPERIMENTS

This section compares our method with baselines. The following sections provide the implementation details (Sec. B), the main results for performance comparison (Sec. 4.2), and ablation studies conducted to analyze the results in three benchmark datasets (Sec. 4.3). Additional experimental details can be found in the Sec D of the supplementary material. The key findings of this study are as follows: 1) sentence-level hierarchical supervision enhances VL alignment by improving linguistic compositionality (Tab. 2); 2) disentangling components with hierarchical structures leads to better modeling of the inductive biases of sentences (Tab. 3); and 3) compositional structure improves the discrimination of positive and negative pairs to represent descriptive sentences better (Tab. 1).

### 4.1 EXPERIMENTAL SETTINGS

**Implementation details.** We build our method based on GLEE (Wu et al., 2024), a pre-trained foundation model composed of MaskDINO (Li et al., 2023a) and CLIP (Radford et al., 2021b) text-image encoders. GLEE was selected as a baseline because, despite being a powerful vision–language foundation model in many benchmarks (e.g., RefCOCO (Yu et al., 2016)), it still faces challenges in contextualizing text embeddings. This study demonstrates that a lightweight hierarchy entailment mechanism can address this limitation and yield further performance gains. For implementations, we use only the HiVG dataset for training, which contains 10 K hierarchy captions. We provide more details of the experimental settings in Sec. B of the supplementary materials.

**Benchmarks and evaluation metrics.** We evaluate the language-based object detection capabilities in two different benchmarks. 1) $D^3$ (Xie et al., 2023) dataset is a widely used benchmark for visual grounding tasks. The dataset includes negative instances, multi-target scenarios, and long sentences. 2) Omnilabel (Schulter et al., 2023) dataset is an open-vocabulary detection dataset. Omnilabel provides an evaluation of compositionality from perspectives such as spatial relationships, actions, and numeracy within referring objects. We perform mean average precision (mAP), a standard evaluation metric, to validate the language-based object detection task.

### 4.2 MAIN RESULTS

We investigate the impact of object detection on disentanglement and hierarchical representation learning through a set of research questions.

| Model | Backbone | D³ (default) | | | D³ (length) | | | | OmniLabel (default) | | | Omnilabel (length) | | |
|---|---|---|---|---|---|---|---|---|---|---|---|---|---|---|
| | | Full | Pres | Abs | S | M | L | XL | AP | $AP_c$ | $AP_d$ | S | M | L |
| OFA-L (Wang et al., 2022) | RN50 | 4.2 | 4.1 | 4.6 | 4.9 | 5.4 | 3.0 | 2.1 | 2.7 | 2.7 | 2.6 | 3.6 | 2.7 | 2.3 |
| MDETR (Kamath et al., 2021) | RN101 | - | - | - | - | - | - | - | - | 4.7 | 9.1 | 6.4 | 4.6 | 4.0 |
| OWL (Minderer et al., 2022b) | ViT-B | 9.6 | 10.7 | 6.4 | 20.7 | 9.4 | 6.0 | 5.3 | 8.0 | 15.6 | 5.4 | 5.7 | 5.4 | 6.2 |
| UNINEXT (Lin et al., 2023) | RN50 | 21.6 | 23.7 | 15.4 | 23.6 | 22.6 | 20.5 | 18.4 | 22.2 | 27.2 | 18.8 | - | - | - |
| G-DINO (Liu et al., 2024a) | Swin-T | 20.7 | 20.1 | 22.5 | 22.6 | 22.5 | 18.9 | 16.5 | 19.3 | 23.6 | 16.4 | 29.4 | 14.8 | 8.2 |
| GEN (Zhao et al., 2024) | Swin-T | 21.4 | 20.6 | 23.7 | 28.1 | 24.5 | 17.4 | 11.5 | 22.2 | 27.2 | 18.8 | - | - | - |
| GLIP (Li et al., 2022b) | Swin-T | 19.1 | 18.3 | 21.5 | 22.4 | 22.0 | 16.6 | 10.6 | 19.3 | 23.6 | 16.4 | 29.4 | 14.8 | 8.2 |
| GLEE-Lite* (Wu et al., 2024) | RN50 | 27.6 | 26.8 | 30.1 | 30.0 | 27.6 | 26.9 | 17.2 | 21.7 | 36.6 | 15.4 | 28.4 | 13.8 | 10.3 |
| GLIP + DesCo (Li et al., 2023b) | Swin-T | 24.2 | 22.9 | 27.8 | 24.3 | 21.9 | 16.4 | 11.5 | 23.8 | 27.4 | 21.0 | 33.7 | 19.0 | 13.7 |
| GLEE-Lite-Scale + DesCo | RN50 | 28.3 | 27.6 | 30.3 | 30.2 | 28.4 | 27.8 | 18.2 | 24.6 | 37.3 | 18.3 | 32.0 | 17.0 | 13.2 |
| GLEE-Lite-Scale + TaSe | RN50 | 30.7 | 29.9 | 33.2 | 31.8 | 31.2 | 30.3 | 19.8 | 26.9 | 36.8 | 21.2 | 33.1 | 19.3 | 14.8 |

Table 1: Evaluation on D³ (Xie et al., 2023) and OmniLabel (Schulter et al., 2023). D³ provides three types of descriptions: absence (ABS), presence (PRES), and full (FULL). text length. For OmniLabel, the final AP is computed as the geometric mean of category-level ($AP_c$) and description-level ($AP_d$) scores. Note that the evaluation results of GLEE-Lite-Scale* are reproduced.

| | D³ | | | OmniLabel | | |
|---|---|---|---|---|---|---|
| | FULL | PRES | ABS | AP | $AP_c$ | $AP_d$ |
| Original GLEE | 27.6 | 27.1 | 30.5 | 21.7 | 36.6 | 15.4 |
| + LoRA (base) | 27.5 | 26.7 | 30.0 | 21.7 | 36.5 | 15.5 |
| + $\mathcal{L}_{CL}$ | 26.9 | 26.1 | 29.1 | 23.9 | 36.7 | 17.7 |
| + $\mathcal{L}_{RE}$ | 27.5 | 26.7 | 30.0 | 25.7 | 36.9 | **19.8** |
| + $\mathcal{L}_H$ (ours) | **28.6** | **27.8** | **31.6** | **26.2** | **38.9** | 19.2 |
| ⌐ w/ $\mathcal{L}_{H+}$ | 28.8 | 27.7 | 31.8 | 24.8 | 37.1 | 18.5 |
| ⌐ w/ $\mathcal{L}_{H-}$ | 27.7 | 27.0 | 30.1 | 23.6 | 36.4 | 18.4 |
| + Reverse $\mathcal{L}_H$ | 26.7 | 25.8 | 29.3 | 22.1 | 36.8 | 15.3 |
| + TriDe module | 28.8 | 27.5 | 32.9 | 25.8 | 35.6 | 20.3 |

Table 2: GLEE trained with hierarchy entailment. $\mathcal{L}_{CL}$ and $\mathcal{L}_{RE}$ represnt contrastive loss (Oord et al., 2018) and RE embedding objective, respectively.

| | D³ | OmniLabel |
|---|---|---|
| Where-to-apply disentanglement | | |
| w/o disentangling | 27.8 (+1.0) | 26.2 (+4.5) |
| Token-level disentangling | **30.7** (+3.1) | **26.9** (+5.2) |
| ⌐ Identity initialization | 30.7 (+3.1) | 26.9 (+5.2) |
| ⌐ Uniform initialization | 28.8 (+1.2) | 26.4 (+4.4) |
| After pooling | 28.6 (+1.4) | 22.6 (+0.9) |
| How-to-apply disentanglement | | |
| Self-attention | 29.0 (+1.4) | 25.5 (+3.8) |
| Learnable query | 29.6 (+2.0) | 24.7 (+3.0) |
| Learnable key & value | **30.7** (+3.1) | **26.9** (+5.2) |
| Effectiveness of disentangling components (# of learnable vector) | | |
| 1 (w/o disentanglement) | 29.4 (+1.8) | 25.4 (+3.7) |
| 2 (Object + Attribute) | 29.5 (+1.9) | 26.5 (+4.8) |
| 3 (Object + Attribute + Relation) | **30.7** (+3.1) | **26.9** (+5.2) |

Table 3: Comparison between disentangled representations with hierarchy entailment

**Does learning hierarchical entailment improve generalization?** As shown in Tab. 1, the proposed model improves upon the baseline by fine-tuning only the LoRA and TriDe. Compared to GLEE, which served as the vision foundation model, we observe improvements of +3.1 in D³ and +5.2 in Omnilabel AP scores. The AP scores in OmniLabel show that hierarchical learning improves performance in zero-shot evaluation, and the gains observed on open-vocabulary benchmarks further demonstrate its effectiveness.

**Does hierarchical learning provide greater benefits than caption augmentation like DesCo?** We further evaluate the performance of GLEE with caption augmentation based on DesCo (Li et al., 2023b). To apply this augmentation, we randomly sample from HiVG. The selected sentence is concatenated with the original caption, and the augmented components are pooled separately and then averaged. The DesCo improves the GLEE model +2.4 AP on D³ and +2.3 on Omnilabel. While caption augmentation increases textual diversity, our hierarchy learning further enhances `TaSe` by enabling accurate distinction of positives and negatives, even when sentences share category names or attributes.

**Qualitative results.** We present two qualitative examples in Fig. 7 to illustrate the effectiveness of our hierarchical entailment learning. The first case shows negative cases containing an attribute (i.e., blue), and the second case presents a positive case with attributes and descriptive relations (e.g., numeracy and text in the image). In the first case, GLEE incorrectly assigns a confident score of 0.92 to the language query. In the second case, GLEE predicts all bikes as positives, including those that do not correspond to the queried bike. On the other hand, `TaSe` captures contextual information related to category names, and hierarchical entailment helps reduce false positives. More qualitative results are provided in Figs. 16 and 17 of the supplementary material.

## 4.3 ABLATION STUDIES

**In hierarchy entailment loss, is it better to learn positives or negatives?** We ablate the hierarchy entailment loss to compare learning with positive and negative pairs. Fine-tuning with positives im-

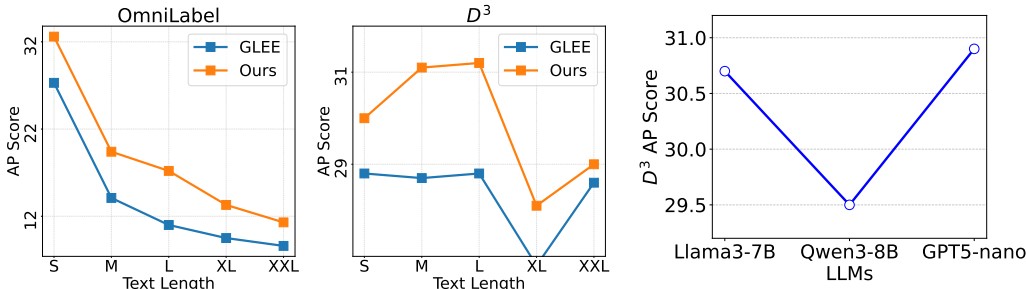

Figure 5: Performance analysis of Omnilabel and $D^3$ based on sentence length. S: [0, 3], M: [4, 7], L: [7, 10], XL: [10, 13], and XXL: [13, $\infty$].

Figure 6: Evaluation of how captions generated by different LLMs affect object detection performance

proves OmniLabel by $+3.1$ AP, while negatives yield $+1.9$ AP. Combining both provides the best generalization in zero-shot settings. To validate this, we invert the objective and observe performance degradation when positives and negatives are aligned in opposite directions (Tab. 2). These findings highlight the role of hierarchical entailment in building effective sentence-level embeddings and suggest that aligning semantically meaningful sentences with visual representations improves performance.

**What advantages does our hierarchical loss offer over traditional contrastive loss?** In Tab. 2, we conduct an ablation study to validate the effectiveness of our hierarchical loss. Within the base setting (GLEE with LoRA), we evaluate three configurations: (1) conventional contrastive loss ($\mathcal{L}_{CL}$); (2) the RE objective ($\mathcal{L}_{RE}$); and (3) the proposed method. For sentence-level hierarchy aggregation, our loss $\mathcal{L}_H$ outperforms contrastive baselines. Conventional contrastive learning causes embeddings of identical category names to diverge when descriptive information differs, whereas our reference-based hierarchy induction aligns them hierarchically and improves sentence-level meaning and performance.

**Ablation on where and how to disentangle in text representation.** Tab. 3 reports ablation studies analyzing the design choices of the TriDe module. Interestingly, we observe that *where* text embeddings are disentangled has the greatest influence on learning granularity. We compare three modes for constructing compositional text embeddings: (1) no disentanglement, (2) token-level disentanglement, and (3) disentangled text embeddings after pooling. Pooling compresses information and limits effective disentanglement, while the no-disentanglement approach is insufficient for capturing sentence-level contextualization. Token-level disentanglement generalizes better and yields the best performance, with module initialization also having a substantial impact on the results. In exploring *how* to design the TriDe module, we investigate disentanglement under three self-attention variants: direct text alignment, learnable queries, and key–value configurations. Key–value attention outperforms query-only and self-attention mechanisms. Key (indexing)–value (content) attention preserves independent subspaces and yields more structured semantic features than query-based approaches. We provide the disentangled embedding results in supplementary material, Fig. 15.

**Is it beneficial to disentangle the representation into three components?** Conventional detectors (Li et al., 2023b; Yuksekgonul et al., 2022) disentangle objects and attributes, whereas we separate representations into three components—object, attribute, and relation—and evaluate their effectiveness. As shown at the bottom of Tab. 3, overall, the performance of three-component disentanglement is higher than two-component (i.e., object and attribute) disentanglement. These findings suggest that three-component disentanglement introduces an inductive bias for complex linguistic structures, making longer sentences more robust to negatives. The granularity of text embeddings reveals features that characterize their representational properties. To further disentangle these components, explicit criteria for dataset composition are required.

**Performance comparison by sentence length.** To evaluate whether modeling contextualization features using O-A-R components affects performance as a function of sentence length, we evaluate results across five length intervals [0, $\infty$]. As shown in the Fig. 5, performance improvements are observed in all intervals, with a large gain in the [7, 10] interval. This higher proportion of captions

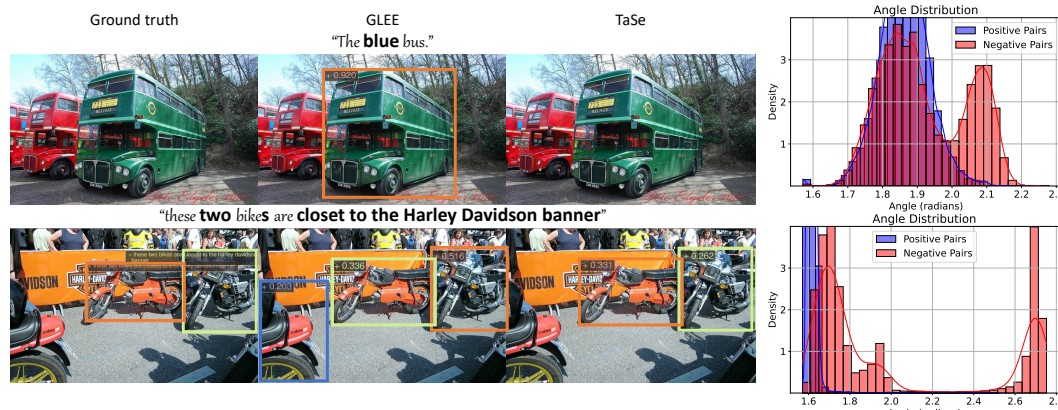

Figure 7: Qualitative analysis on Omnilabel data (Schulter et al., 2023). We visualize and compare the results between our baseline (GLEE) and TaSe. We select the scenario that includes attributes and relations for referring to a category name.

Figure 8: Comparison of exterior angles between GLEE and TaSe

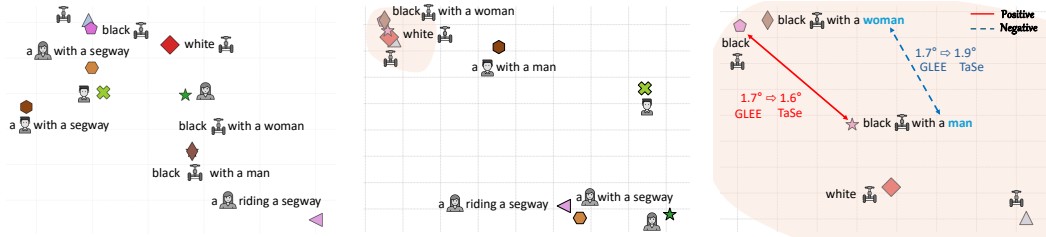

(a) GLEE embedding visualization    (b) TaSe embedding visualization    (c) Zoom in TaSe embedding

Figure 9: Comparison between GLEE and TaSe text embedding. We set objects to correspond to each icon. Our proposed hierarchical representation learning aligns text embedding of GLEE.

is consistent with the HiVG data statistics. This result aligns with the higher proportion of captions in this interval in the HiVG dataset. Statistical data are in Supplementary Fig. 12.

**Performance Comparison based on Captions Generated by Different LLMs.** We constructed a caption dataset containing positive and negative examples using three LLMs: Llama3-8B (Dubey et al., 2024), Qwen3-8B (Yang et al., 2025), and GPT5-nano (OpenAI, 2025). For each sample in Visual Genome (Krishna et al., 2017), we generate 10 K captions. To reduce hallucination, we enforced the hierarchical format "Tier 3 ⇒ Tier 2 ⇒ Tier 1" using the delimiter "⇒". Captions that did not conform to this structure are filtered and regenerated. As shown in the Fig. 6, GPT-5-nano data demonstrate the highest performance, with Llama second and Qwen third.

**How are embeddings structured after disentanglement?** While a few negative pairs still exhibit large angles, Fig. 8 confirms that positive and negative pairs are effectively aligned in the embedding space. To examine whether this alignment follows the intended structure after disentanglement and hierarchical aggregation, we visualize the t-SNE projection of the trained TaSe embeddings. As shown in Fig. 9a, GLEE is dispersed around category names, whereas TaSe realigns embeddings around objects and preserves robust angular distance for negatives corresponding to attributes or relations. This is evident in the "segway" object, where the captions "black segway with a woman" and "black segway with a man" lie at different angles from the reference point "black segway."

## 5 CONCLUSION

This study proposed a disentanglement and hierarchy aggregation framework for constructing contextualized sentence representations within language-based object detection. Additionally, we generate re-captioned data for object detection using hierarchical concepts. TaSe improved the linguistic compositionality, which serves as a key learning factor and leads to competitive results. The results indicated that hierarchy entailment allows learning the granularity of text embedding to distinguish descriptive sentences. This study highlights the need for further exploration of the underlying linguistic compositionality in future studies for downstream vision tasks.

ETHICS STATEMENT

During the preparation of this work, the author used ChatGPT (Hurst et al., 2024) in order to improve readability. After using this tool/service, the author reviewed and edited the content as needed and takes full responsibility for the content of the publication.

REPRODUCIBILITY STATEMENT

We recaptioned our dataset using the publicly available Llama 3 (Dubey et al., 2024) and Qwen 3 (Yang et al., 2025) released on the Hugging Face Hub (Wolf et al., 2020). GPT-5 (OpenAI, 2025) was accessed via the OpenAI API. Additional statistics and details of the dataset are presented in Sec. A of the supplementary material. The code for the experiments is available in the supplementary material.

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

## A  DETAILS OF HiVG

### A.1  CREATING HIERARCHICAL POSITIVE CAPTIONS

Our hierarchical captioning pipeline is illustrated in Tab. 4, which highlights key differences from conventional captioning approaches. We generate a 10 K re-captioned dataset from the Visual Genome dataset. We first filter out all Visual Genome captions with fewer than six words to ensure sufficient semantic richness. For positive captions, we leverage in-context learning using LLaMA (Dubey et al., 2024) to transform the remaining Visual Genome captions into a three-tier hierarchical structure. To enhance attribute diversity within the captions, we draw on common visual concepts (Huang et al., 2023; Lin et al., 2024) to define a set of visual attributes (spatial, color, number, and size) and randomly select one to modify the object for alternative object attributes. Based on our experiments, we set the randomization ratio to 50%. To fully align with the in-context learning demonstration format, samples that lack attributes or relations are also incorporated into the learning process, following the approach suggested by Min et al. (2022).

### A.2  CREATING HIERARCHICAL NEGATIVE CAPTIONS

One of the challenges in language-based object detection is effectively handling negative samples, which often report higher false negative rates compared to false positives. To address the issue, we focus on both re-captioning hard and easy negative samples. For hard negative samples in tier 1, we replace the positive object with an antonym (e.g., *man* is replaced by *woman*) or a random concrete noun. Easy negative samples are generated by selecting nouns from ImageNet1000 (Deng et al., 2009) classes and lexical databases such as WordNet and ConceptNet. Additionally, we insert a negative determiner to the object (e.g., *dog* is switched to *no dog*).

In tier 2, we reuse the same set of visual attributes from the positive captions but replace them with semantically different attributes (e.g., *tall building* is replaced by *short building*) for generating hard negative samples. We use LLM-based mask-filling (Liu et al., 2019) to diversify attributes by substituting them with contextually plausible but semantically different terms or by prepending "not" to create hard negatives (e.g., tall building → not tall building).

In tier 3, we use a set of common spatial relations (e.g., above and beside) and object-specific relations from the Visual Genome dataset. These pre-defined object-specific relations ensure that the relation is contextually relevant to the object in question. To introduce hard negatives, we apply absence-based transformations by replacing affirmative relations with their negative counterparts (e.g., *with* is replaced by *without*). We also leverage LLaMA's (Dubey et al., 2024) sentence completion capability to generate further relation diversity.

Captions that do not adhere to the hierarchical structure are filtered out. By re-captioning using a multi-tiered set of positive and negative captions, our approach is intended to facilitate the learning of hierarchical representations, thereby improving linguistic compositionality.

### A.3  HiVG STATISTIC ANALYSIS

The combined dataset consists of 286,006 annotations, with the majority containing $8.75 \pm 1.34$ words, as shown in Fig. 12. Caption length distributions across Positive and Negative samples are largely consistent within each tier. Tier 3 contains the most linguistically diverse and structurally rich captions, which may be particularly beneficial for semantic reasoning.

We report that our hierarchy dataset, HiVG, in Fig. 13. Leveraging Visual Genome data for re-captioning, we create a more diverse dataset by incorporating a wider range of classes and the LLM and other datasets.

### A.4  HUMAN EVALUATION FOR GENERATED CAPTIONS

To validate the quality of the proposed HiVG dataset, we conducted a human evaluation. Specifically, we aimed to assess whether the three-tier hierarchies in the LLM-generated captions accurately reflect our intended design. To this end, we asked 50 participants to verify whether the captions in each tier describe the same underlying objects. The caption sets were randomly assigned to partici-

| Dataset / Approach | Positive Captions | Negative Captions | Entailment Structure |
|---|---|---|---|
| Visual Genome (Krishna et al., 2017) | Flat object-centric region descriptions | - | - |
| Image Paragraphs (Krause et al., 2017) | Multi-sentence paragraphs per image | - | Narrative-level cohesion only |
| HierarCaps (Alper & Averbuch-Elor, 2024) | LLM-generated hierarchical captions | LLM&NLI-based structure contradiction samples | Inferred via entailment prediction |
| HiVG | Explicit object → attribute → relation chains, used in in-context learning | WordNet-informed hard negatives and ImageNet-based categories | Explicit tiered entailment |

Table 4: Comparison of positive/negative caption strategies and entailment assumptions across datasets. Our method introduces grounded, logic-consistent supervision with object-level structure, unlike prior captioning datasets.

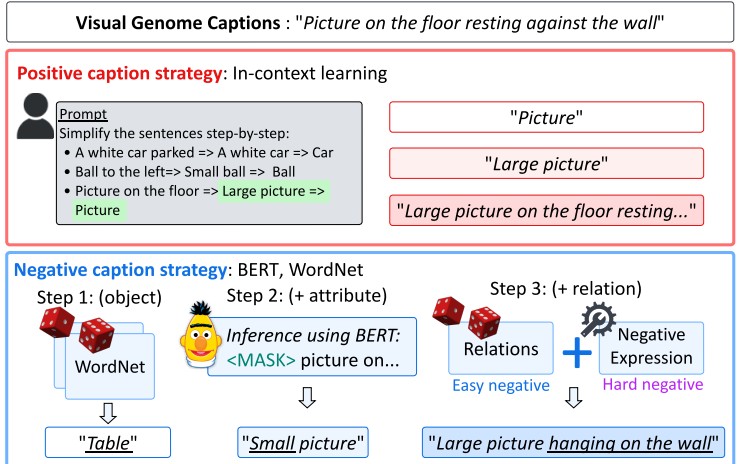

Figure 10: Overview of generating positive and negative captions. Positive captions are derived using in-context learning based on Llama3 (Dubey et al., 2024). We transform Visual Genome captions into structured forms: object (category name), attribute (category name with an attribute), and relation (category name with an attribute and a relation). Negative captions are constructed through a multi-step process: 1) retrieving antonyms or random concrete nouns from lexical databases for negative objects; 2) using LLM-based mask-filling combined with pre-defined visual attributes to generate semantically different negative attributes; and 3) using pre-defined object-specific relations to create negative relations.

pants, and a total of 500 samples were evaluated. A screenshot of the evaluation interface is provided in Fig. 14.

The quality of the proposed HiVG dataset is remarkably high, supporting the effectiveness of our data-generation pipeline. 98.8% of the evaluated samples follow the intended hierarchical design, with only 6 out of 500 samples depicting different target objects. The six failure cases are listed below:

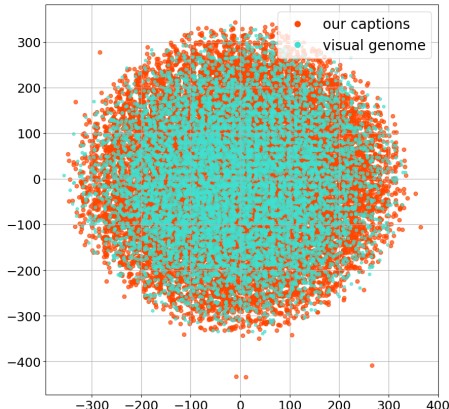

Figure 11: t-SNE plot comparing the clustering of our captions with Visual Genome captions. Green and orange represent original Visual Genome captions and our re-captioned dataset, respectively. Our captions exhibit a broader spread in the 2D space, indicating greater semantic diversity compared to the original Visual Genome captions.

---

**Failure case 1**
**Tier 1:** pen
**Tier 2:** a white pen
**Tier 3:** papers and a white pen on top of tab

**Failure case 2**
**Tier 1:** cpu
**Tier 2:** A black cpu unit
**Tier 3:** A black cpu unit unit on the floor under a computer monitor that is on

**Failure case 3**
**Tier 1:** man
**Tier 2:** a black man
**Tier 3:** a bartender wearing a mustache and a tie

**Failure case 4**
**Tier 1:** boat
**Tier 2:** the yellow boat
**Tier 3:** the yellow boat masts in the water

**Failure case 5**
**Tier 1:** woman
**Tier 2:** woman and little girl
**Tier 3:** woman and little girl reading on a beige couch

**Failure case 6**
**Tier 1:** numbers
**Tier 2:** red numbers
**Tier 3:** red numbers 2008 on the side of the buidling

---

## B EXPERIMENTAL SETUP

**Baselines** This paper compares the language-based object detection models on OFA (Wang et al., 2022), OWL (Minderer et al., 2022b), G-DINO (Liu et al., 2024a), GLIP (Li et al., 2022b), UNINEXT (Lin et al., 2023), Desco (Li et al., 2023b), GLIP-GEN (Zhao et al., 2024), and GLEE (Wu et al., 2024).

| Params. | Value |
|---|---|
| Batch size | 4 |
| Optimizer | AdamW |
| Optimizer momentum | $\beta_1 = 0.9, \beta_2 = 0.999$ |
| Rank of LoRA | 16 |
| scaling factor of LoRA | 16 |
| learning rate of LoRA | 5e-6 |
| learning rate of TriDe | 1e-4 |
| Input resolution | $800 \times 800$ |
| loss of class ($\mathcal{L}_{class}$) | 4.0 |
| loss of bbox ($\mathcal{L}_{bbox}$) | 5.0 |
| loss of gIoU ($\mathcal{L}_{giou}$) | 2.0 |
| loss of TaSe ($\mathcal{L}_{TaSe}$) | 5.0 |
| $\lambda$ | 0.1 |

Table 5: Hyperparameters setting

| Layers | # of Params. (M) |
|---|---|
| Image backbone | 23.5 |
| Text encoder | 126.3 |
| Detector | 31.5 |
| LoRA | 0.4 |
| TriDe | 1.9 |
| VL | 3.2 |
| Trainable params. | 5.4 (2.93%) |

Table 6: Model configuration.

### B.1 IMPLEMENTATION DETAILS

We sample 16 images per batch and further select 6 corresponding sentences per image for hierarchy learning. We employ AdamW (Loshchilov & Hutter, 2017) to optimize the trainable model, using a learning rate of $1 \times 10^{-4}$ for the TriDe module and $5 \times 10^{-6}$ for LoRA. For the comparison with baselines, our detector was trained for 60 K iterations, the same as in the ablation studies. Following Alper & Averbuch-Elor (2024), the RE loss was set with a positive-to-negative ratio of 10:4. In case of $\mathcal{L}_{\text{H}}$, we conduct experiments wit positive-to-negative ratio of 2:1. The values of $\gamma$ is set to 0.1.

### B.2 MODEL SIZE AND BUDGET

For fine-tuning the pre-trained GLEE, we only train LoRA layers, TriDe module, and VL fusion layers. We train a total of 5,447,680 parameters, which is an efficient approach that reduces memory usage by 2.93% of the model parameters. On average, 10 iterations are executed, requiring 62.2 ms per inference. The disentangling and aggregation step adds an additional 1.57 ms (2.3%) overhead. Experiments were conducted using 4 NVIDIA A6000 GPUs for model training.

**Additional qualitative results.** To evaluate whether our model effectively learns sentence-level hierarchy, we compared its performance with baselines using scenarios including objects, attributes, and relations from the benchmark dataset. As shown in Fig. 16 and Fig. 17, we visualize the results of two scenarios containing an absent example. Given that sentences become longer, many VLMs focus on specific words, such as "running" to detect objects. For example, our model improves performance by capturing richer semantic information, such as the attribute "pink," and understanding contextual meaning, like recognizing the "girl" as the subject of "running." In contrast, our model demonstrates greater robustness in detecting complex relations and predicts bounding boxes more accurately by better understanding object states and relative information.

## C DETAILS OF COMPOSITIONAL LEARNING

We provide the details of the disentanglement modes employed for compositional learning as shown in Alg. 1. The first mode adopts traditional mean-pooling and uses the resulting representation for contrastive learning. The second and third modes involve disentanglement via a TriDe module, followed by contrastive learning based on the aggregated compositional embedding. Specifically, the second mode applies the TriDe module at the token level to leverage information across all tokens, whereas the third mode applies the module after pooling, focusing on sentence-level semantics.

## D ADDITIONAL EXPERIMENTAL RESULTS

---

**Algorithm 1:** Pseudo code for disentangling and hierarchical aggregating paradigm

---

**Input:** Image-text pair $(v_i, t_i)_{i=1}^{B}$; Text encoder with LoRA $\mathcal{T}_\theta$; Learnable vectors
$\mathbf{V_O}, \mathbf{V_A}, \mathbf{V_R}$; Projection embedding $\delta$; Vision backbone $\mathcal{V}_\psi$; MaskDINO $f_\psi$
**Output:** Bounding box and class $Y$; Total Loss $\mathcal{L}$;

**for** *each training iteration* **do**
$\quad \mathbf{X_v} = \mathcal{V}_\psi(v)$
$\quad \mathbf{X} = \mathcal{T}_\theta(t) \cdot \delta$
$\quad$ // TriDe: token-level disentangling
$\quad \mathbf{X} = \mathbf{FFN}(\mathbf{X})$
$\quad \mathbf{O} = \text{CrossAttn}(\mathbf{X}, \mathbf{V_O})$
$\quad \mathbf{A} = \text{CrossAttn}(\mathbf{X}, \mathbf{V_A})$
$\quad \mathbf{R} = \text{CrossAttn}(\mathbf{X}, \mathbf{V_R})$
$\quad \mathbf{E} = \text{Pool}(\mathbf{FFN}(\mathbf{O} + \mathbf{A} + \mathbf{R}))$
$\quad$ //
$\quad \mathbf{Y}, \mathcal{L} = f_\psi(\mathbf{X_v}, \mathbf{E})$
Update $\theta$ by minimize $\mathcal{L}$

| Methods | Visual Encoder | Textual Encoder | val | | testA | | testB | |
|---|---|---|---|---|---|---|---|---|
| | | | Pr@0.5 | N-acc. | Pr@0.5 | N-acc. | Pr@0.5 | N-acc. |
| MCN Luo et al. (2020) | DarkNet-53 | GRU | 28.0 | 30.6 | 32.3 | 32.0 | 26.8 | 30.3 |
| VLT Ding et al. (2021) | DarkNet-53 | GRU | 36.6 | 35.2 | 40.2 | 34.1 | 30.2 | 32.5 |
| MDETR | ResNet-101 | RoBERTa | 42.7 | 36.3 | 50.0 | 34.5 | 36.5 | 31.0 |
| UNINEXT[†] Lin et al. (2023) | ResNet-50 | BERT | 58.2 | 50.6 | 46.4 | 49.3 | 42.9 | 48.2 |
| GLEE (Wu et al., 2024) | RN50 | CLIP | 66.7 | 62.4 | 73.0 | 64.1 | 61.8 | 51.9 |
| GLEE + TaSe | RN50 | CLIP | **69.6** | **69.9** | **76.6** | **69.6** | 67.7 | **58.9** |

Table 7: Zero-shot evaluation results on gRefCOCO (He et al., 2023). Following He et al. (2023), Pr@0.5 (F1=1, IoU$\geq$0.5) denotes a true positive when the predicted bounding box achieves an IoU greater than 0.5 with the ground truth. N-acc. represent positive rate (TN / Positive).

## D.1 ZERO-SHOT GENERALIZATION ANALYSIS

We evaluate zero-shot generalization performance on real-world detection tasks. We compare `TaSe` with baseline models on the gRefCOCO (He et al., 2023) and ODinW datasets (Li et al., 2022b) in Tables 7 and 8. On the ODinW dataset, our model achieved an average improvement of +4.9 AP. On the gRefCOCO dataset, following He et al. (2023), `TaSe` demonstrates improvements in object detection performance with textual queries.

## D.2 ADDITIONAL ABLATION STUIDES

We conducted an ablation study on the model parameters. We compared and analyzed the effects of adjusting the LoRA (Hu et al., 2021) rank (see Fig. 18), the number of tokens (see Fig. 19, and the margin parameter (see Fig. 20) for the disentangled loss.

## E COMPARATIVE ANALYSIS OF TEXT EMBEDDINGS ACROSS STATE-OF-THE-ART VISION-LANGUAGE MODELS

VLMs are pre-trained for text–image alignment, and Fig. 1b shows that they often demonstrate bag-of-words behavior. Recent VLMs incorporate approaches such as MAE-based architectures (Fang et al., 2024) and LLM-connected visual encoders (Jiang et al., 2024) to improve representation quality. We investigate whether the text embeddings of these models improve object detection. For this analysis, we visualize text embeddings from SigLIP (Zhai et al., 2023) and E5-V (Jiang et al., 2024) using the same example shown in Fig. 22a.

| | Pascal VOC | Aerial Drone | Aqua rium | Rab bits | Ego Hands | Mush rooms | Pack ages | Rac coon | Shell fish | Veh icles | Pis tols | Pot hole | Ther mal | Avg |
|---|---|---|---|---|---|---|---|---|---|---|---|---|---|---|
| GLEE | 61.2 | 5.0 | 23.9 | 71.9 | 46.2 | **57.8** | 25.6 | **56.8** | 33.1 | 60.6 | 57.1 | **25.3** | **52.5** | 44.4 |
| TaSe | **61.3** | **14.4** | **24.8** | **81.4** | **54.5** | 20.8 | **63.1** | 50.6 | **41.0** | **60.8** | **61.5** | 18.9 | 46.7 | **49.3** |

Table 8: Zero-shot evaluation results across 13 datasets in ODinW (Li et al., 2022a).

Fig. 22a shows that SigLIP demonstrates embeddings that closely align across distinct objects (e.g., black segway with a man and black segway with a woman), similar to CLIP. For E5-V, the LLM-based embeddings were more object-centered (e.g., woman), yet white segway and black segway remained closer to each other than black segway with a man. This observation indicates that even advanced VLMs can retain ambiguities in object-level representation, as they rely on textual embedding similarity to correlate visual content.

# F  QUALITATIVE RESULTS

## F.1  HIERARCHY TRAINING EMBEDDING ANALYSIS.

We validate the effectiveness of our proposed hierarchical learning approach by visualizing the impact of the angular loss on both inter-tier and intra-tier constraints. For the experimental setup, we randomly initialize 50 two-dimensional embeddings and train them using the original hierarchy loss and our extended loss function.

## F.2  ANALYSIS OF REPRESENTATION DISENTANGLEMENT

To verify whether disentangled embeddings contain distinct embedding representations for each component, we visualize the embedding of each component using t-SNE. For the t-SNE visualization, we construct the embedding space using our HiVG dataset of 10K samples. We then visualize the embeddings based on the language queries that motivated us. As shown in Fig. 15, although there are slight variations across tokens, the embeddings for each component cluster relatively well. This validates that when a sentence is input, each component holds disentangled representations.

## F.3  FAILURE CASES

We present qualitative results that indicate directions for improving the model. As shown in Fig 23, accurately detecting objects remains challenging when additional attributes or relations are included. Natural language contains complex structures, such as negation or compound structures, beyond the O-A-R components. These diverse forms make it difficult to understand finer-grained semantic meaning, and this remains an important research problem. `TaSe` reveals two major failure patterns: i) The model struggles with ambiguous expressions (e.g., looking toward the camera); and ii) Datasets such as RefCOCO contain only a single instance per object category, often resulting in predictions that merge multiple instances into one bounding box.

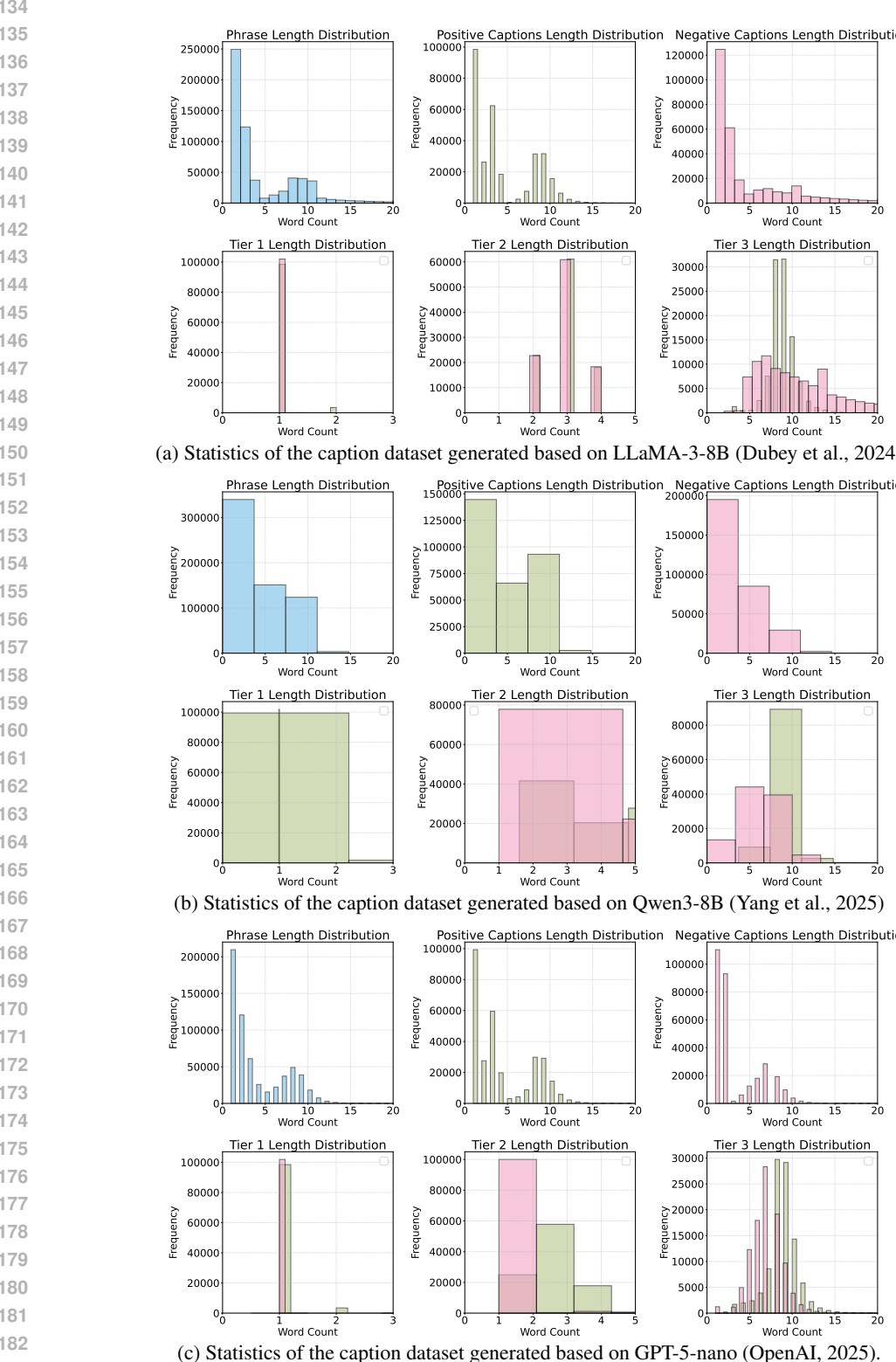

(a) Statistics of the caption dataset generated based on LLaMA-3-8B (Dubey et al., 2024)

(b) Statistics of the caption dataset generated based on Qwen3-8B (Yang et al., 2025)

(c) Statistics of the caption dataset generated based on GPT-5-nano (OpenAI, 2025).

Figure 12: Statistics of HiVG dataset. Top: Distribution of the number of words. (Left) The original Visual Genome dataset. (Middle) positive captions. (Right) Negative captions. Bottom: Distribution of the number of words per tier. (Left) Tier1 - category name. (Middle) Tier2 - attribute. (Right) - Tier3 relation.

| Tier 1 | Tier 2 | Tier 3 | Tier 1 | Tier 2 | Tier 3 |
|---|---|---|---|---|---|
| car | a red car | red car are parked on the side of the road | pen | A red pen | A red writing utensil ontop of a notebook |
| discard | a sports car | a red car parked on the side of a road: BMW 5 Sedan (G30) | bedstr aw | A blank pen | A red pen next to a yellow legal pad with the words 'I don't know what to write about it |

| Tier 1 | Tier 2 | Tier 3 | Tier 1 | Tier 2 | Tier 3 |
|---|---|---|---|---|---|
| table | wooden table | The wooden table and chairs are made of wood | building | Yellow building | Yellow building opposite were the men are standing |
| string orchest ra | summary table | wooden tableware set wooden tableware set dining room chairs | equestri enne | Not yellow building | Yellow building witout the letter l on it |

Figure 13: Re-captioning data examples

## Human Evaluation for Caption Generation

Each question presents a set of images structured in a hierarchical manner:

- **Tier 1: Object only**
  Example: *Segway*

- **Tier 2: Object + Attribute**
  Example: *Black Segway*
  Represents characteristics of the object, such as **size, color, shape, or spatial information**.

- **Tier 3: Object + Attribute + Relation**
  Example: *Black Segway with a man*
  Represents a **relationship** between the object and another entity.

Your task is to determine whether the three tiers describe the **same object** or **different objects**.
Please examine the images carefully and indicate whether they refer to the **same underlying object across tiers**.

---

Does Tier 1, Tier 2, and Tier 3 refer to the same object? *
Tier 1: window
Tier 2: A green window
Tier 3: A green window in the front of the building

○ Same object

○ Different objects

```javascript
function generateMultipleForms() {
  const ss = SpreadsheetApp.openById("");
  const sheet = ss.getSheetByName("");
  const baseFormId = "";

  const values = sheet.getRange(2,5,sheet.getLastRow()-1,1).getValues();
  const texts = values.map(row => row[0]).filter(t => t !== "");

  for (let i = 1; i <= 50; i++) {
    const newFormFile = DriveApp.getFileById(baseFormId).makeCopy(`Survey ${i}`);
    const newForm = FormApp.openById(newFormFile.getId());

    const shuffled = texts.sort(() => Math.random() - 0.5).slice(0, 10);

    newForm.getItems().forEach(item => newForm.deleteItem(item));

    shuffled.forEach(q => {
      const item = newForm.addMultipleChoiceItem();
      item.setTitle(q)
          .setChoices([
            item.createChoice("Same object"),
            item.createChoice("Different objects")
          ])
          .setRequired(false);
    });

    Logger.log(`📌 Created: ${newForm.getTitle()} → ${newForm.getEditUrl()}`);
  }
}
```

Figure 14: Google form-based questionnaire for human evaluation

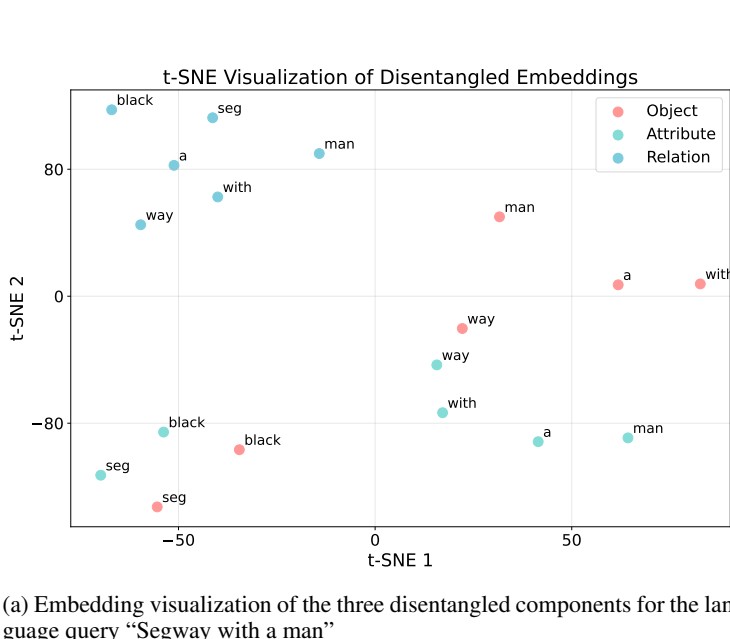

(a) Embedding visualization of the three disentangled components for the language query "Segway with a man"

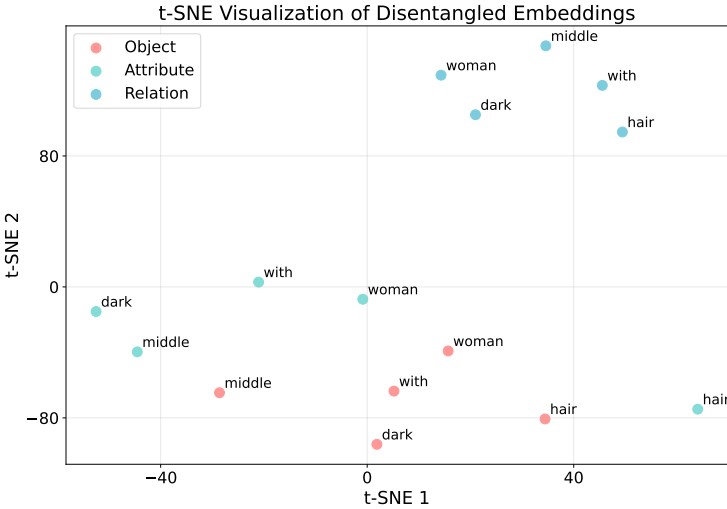

(b) Embedding visualization of the three disentangled components for the language query "middle woman with dark hair"

Figure 15: t-SNE visualization of disentangled text embedding

Figure 16: Qualitative analysis on Omnilabel data Schulter et al. (2023). We visualize and compare the results between GLEE and TaSe. We visualized the prediction results for both positive and negative captions of the same image.

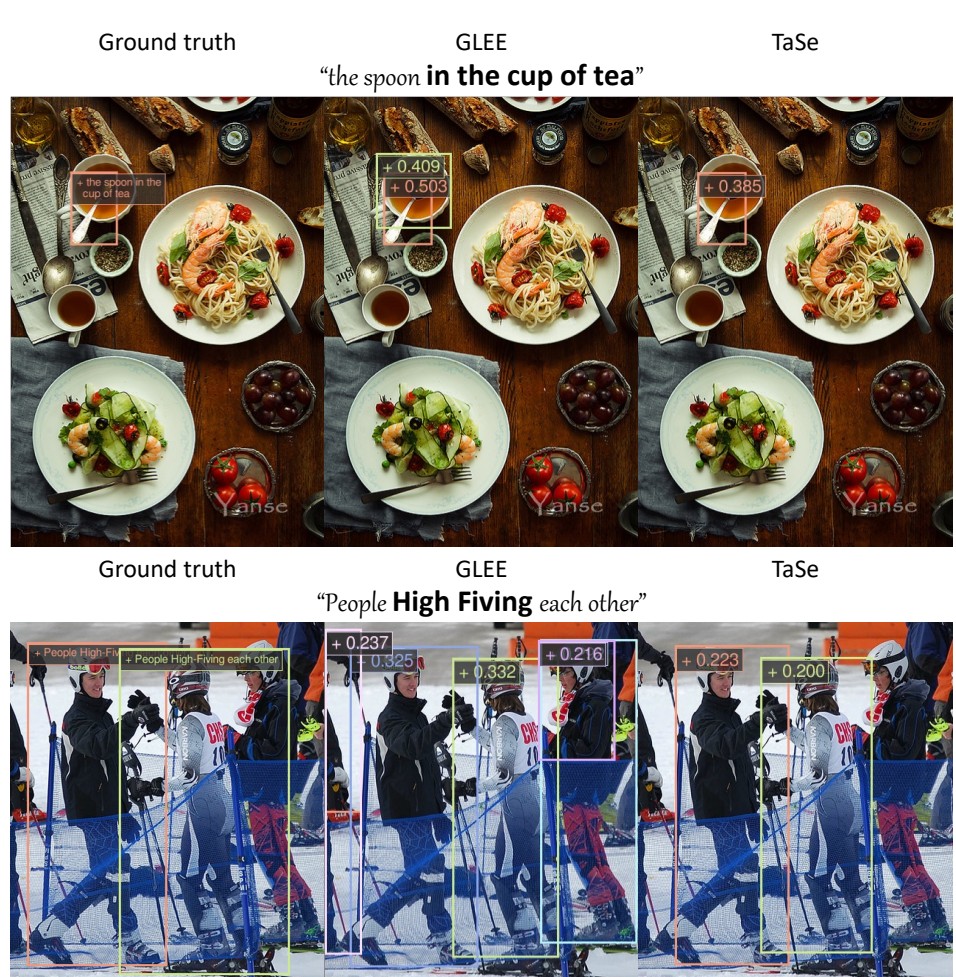

Ground truth  GLEE  TaSe

"*the spoon* **in the cup of tea**"

Ground truth  GLEE  TaSe

"*People* **High Fiving** *each other*"

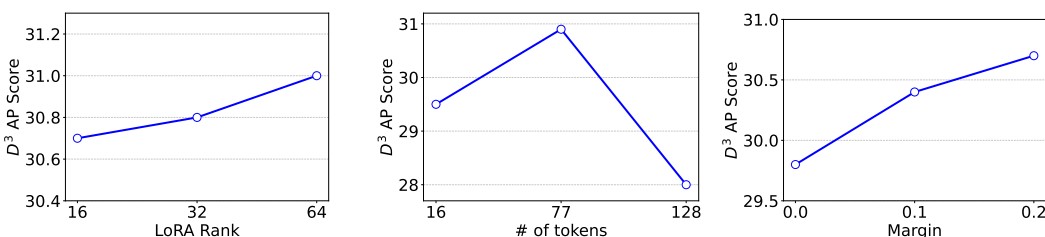

Figure 17: Qualitative analysis on Omnilabel data Schulter et al. (2023). We visualize and compare the results between GLEE and `TaSe`.

Figure 18: Analysis on the effect of different LoRA ranks

Figure 19: Analysis on the number of tokens

Figure 20: Analysis on the margin $m$ in the $\mathcal{L}_{\text{TriDe}}$

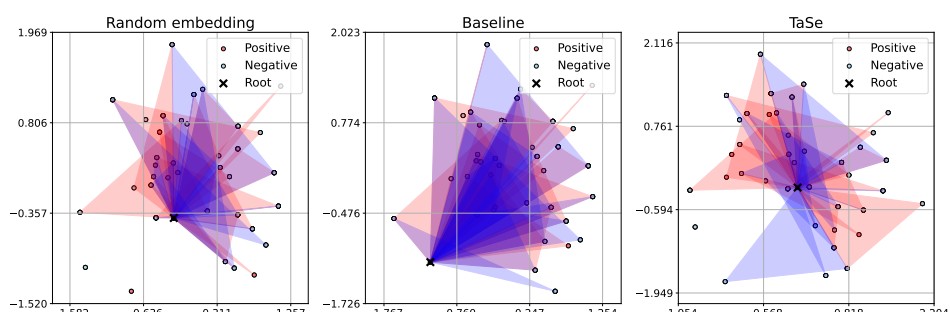

Figure 21: Visualization of angular embeddings (without dynamic reference). Triangles illustrate learned pairs with respect to the root: positive pairs (red) and negative pairs (blue) are connected to depict directional behavior. Positive pairs are expected to align in similar directions from the root, while negative pairs should diverge. While the baseline tends to increase radial distance more than meaningful angular adjustment, our objective function encourages more structured representations guided by directional alignment.

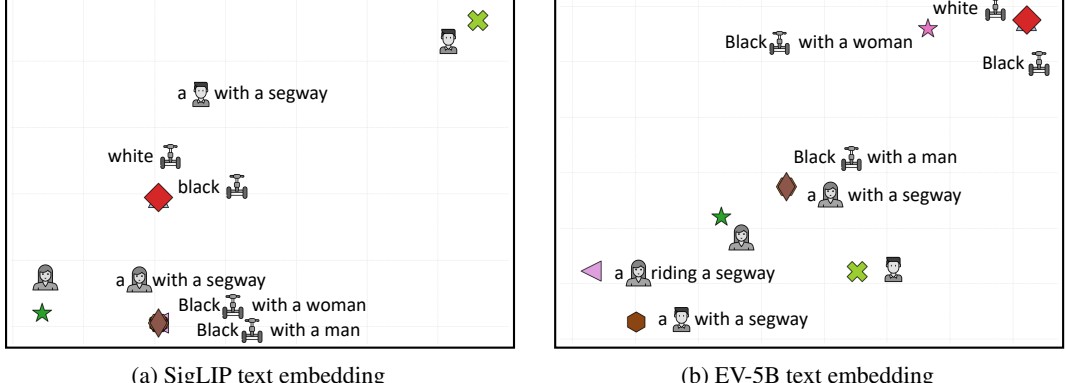

(a) SigLIP text embedding         (b) EV-5B text embedding

Figure 22: t-SNE plot of text embedding between SigLIP (Zhai et al., 2023) and EV-5B Jiang et al. (2024).

Ground truth                    TaSe

"The **Two** people who **are wearing hats on their head.**"

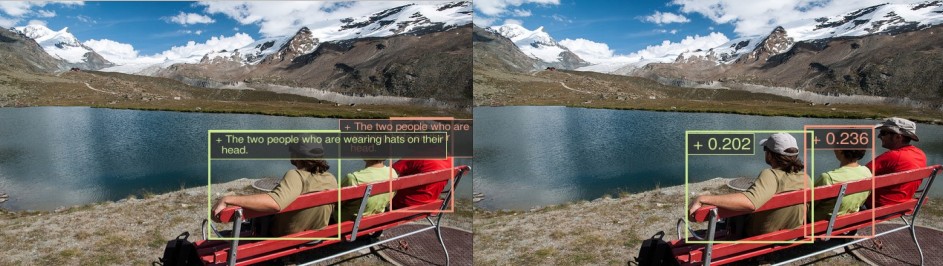

"the **two** vehicles that **don't** have a visible **animated** character **on the side**"

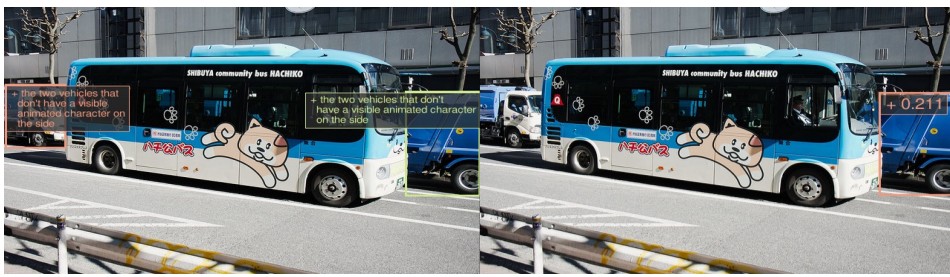

Ground truth                    TaSe

"The sheep that are **looking toward the camera.**"

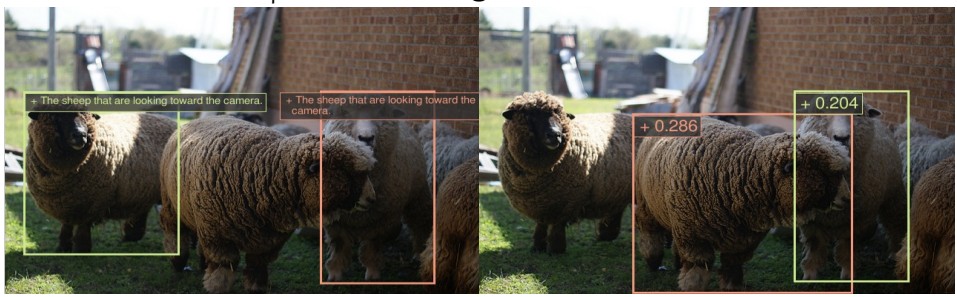

"The wine glasses the people are holding"

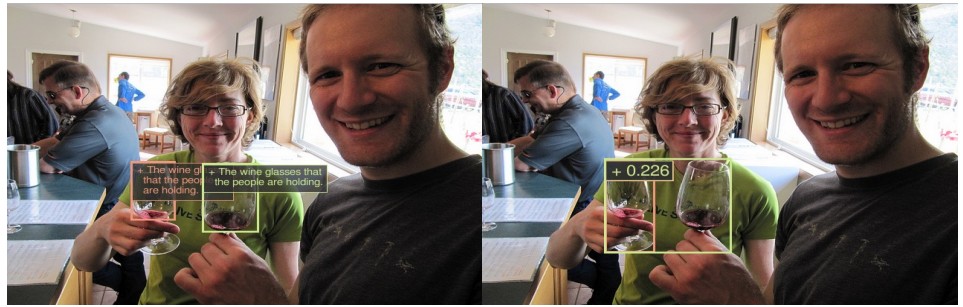

Figure 23: Qualitative results for failure cases.

