# OpenReview forum: "Talk in Pieces, See in Whole: Disentangling and Hierarchical Aggregating Representations for Language-based Object Detection"
_ICLR.cc/2026/Conference — Submitted to ICLR 2026_

### Official Review · Reviewer_MTQ7 · 2025-10-26

**Soundness:** 3
**Presentation:** 2
**Contribution:** 3
**Rating:** 4
**Confidence:** 4

**Summary:**

The paper addresses a key limitation in vision-language models: their poor compositional understanding of complex textual queries in language-based object detection (LBOD). Current models behave like “bags-of-words,” often misinterpreting sentences that combine object names, attributes, and relations. The authors propose TaSe (Talk in Pieces, See in Whole), which consists of:
1. HiVG Dataset: A hierarchical synthetic captioning dataset derived from Visual Genome, organized into object–attribute–relation tiers.
2. TriDe Module: A component-wise disentanglement mechanism that separates text embeddings into object, attribute, and relation subspaces.
3. Hierarchical Aggregation: A new loss based on hierarchical entailment using a Euclidean-space “Radial Embedding (RE)” objective to reconstruct contextualized sentence representations.
TaSe integrates into GLEE (a MaskDINO + CLIP-based VL detector) and achieves substantial gains on OmniLabel and D^3 benchmarks.

**Strengths:**

1. The paper convincingly diagnoses a key weakness of current VL detectors: poor sentence-level compositionality and contextual grounding. The empirical t-SNE analysis in Fig. 1 is effective in illustrating why text embeddings collapse across different relational phrases.
2. Comprehensive experiments on multiple benchmarks, with both quantitative (Tables 1–3) and qualitative (Figs. 5–7  results.
3. The writing is clear and methodically organized.

**Weaknesses:**

1. HiVG, the proposed hierarchical dataset, is central to the paper but only briefly described in the main text. There’s no quantitative analysis of HiVG’s diversity, quality, or balance (e.g., number of distinct attributes/relations, comparison with Visual Genome). Since HiVG is synthetic and LLM-generated, potential hallucination or linguistic bias needs evaluation.
2. Consider including a human evaluation of caption quality or a robustness test (e.g., sensitivity to paraphrases).
3. The framework combines multiple non-trivial modules (TriDe, RE, hierarchical losses). While each part is motivated, the cumulative complexity may limit reproducibility. It’s unclear how much improvement comes from disentanglement vs. better data vs. hierarchical supervision, though ablations partly address this.



Minor Presentation Issues：
1. Some equations (e.g., Eq. 3, 6) are dense and could use more intuition.

2. Figures could label components more clearly (especially in Fig. 2).

**Questions:**

1. The final text embedding E=pooling(O+A+R). What pooling function is used (mean, [CLS] token, attention)? Have you compared different pooling methods to see how they affect compositional sensitivity?
2. How is the “reference point”  r in Eq. 6 chosen or updated?
3. Why were structure-aware or compositional baselines like CLIP-Adapter, LogicZSL, or MDETR not included?
4. Have you identified scenarios where the hierarchical structure fails — e.g., ambiguous relational clauses or overlapping attributes?

---

> ### Author Response · Authors · 2025-11-22
> **Official Comment by Authors**
>
> We thank the reviewer for their thoughtful and constructive feedback. We address the concerns as follows:
>
> > **W1:** HiVG, the proposed hierarchical dataset, is central to the paper but only briefly described in the main text. There’s no quantitative analysis of HiVG’s diversity, quality, or balance (e.g., number of distinct attributes/relations, comparison with Visual Genome). Since HiVG is synthetic and LLM-generated, potential hallucination or linguistic bias needs evaluation.
>
> - Thank you for your valuable suggestion! We have added a t-SNE visualization showing the overall distribution of our re-captioned data in Appendix `Fig. 11`. In addition, we have included comparative distributions for the LLaMA3-based captioning dataset and the two newly added LLM-generated caption datasets in Appendix `Fig. 12`.
>
> > **W2:** Consider including a human evaluation of caption quality or a robustness test (e.g., sensitivity to paraphrases).
>
> - To validate the quality of the proposed HiVG dataset, we conducted a human evaluation. Specifically, we aimed to assess whether the three-tier hierarchies in the LLM-generated captions accurately reflect our intended design.
> - We recruited 50 participants to verify whether the captions in each tier describe the same underlying objects. The caption sets were randomly assigned to participants, and a total of 500 samples were evaluated. As a result, **98.8%** of the evaluated samples adhered to the intended hierarchical design, underscoring the effectiveness of our data-generation pipeline and the quality of the resulting HiVG dataset. Additional details, including the failure cases, are provided in Appendix `Sec. D.2` of the revised manuscript.
>
>
> > **W3:** The framework combines multiple non-trivial modules (TriDe, RE, hierarchical losses). While each part is motivated, the cumulative complexity may limit reproducibility. It’s unclear how much improvement comes from disentanglement vs. better data vs. hierarchical supervision, though ablations partly address this.
>
> - Thank you for your suggestion regarding the component-wise ablation (i.e., examining the significance of hierarchy, disentanglement, and improved data). First, the proposed HiVG dataset is the foundation of our work, as it is explicitly designed with a three-tier hierarchy and tailored to learn hierarchical structure and linguistic disentanglement by design. Without the HiVG dataset, it becomes non-trivial to properly teach hierarchy and disentanglement. In other words, it is difficult to quantitatively disentangle the contribution of the dataset itself from that of the hierarchy/disentanglement learning objectives.
> - To further assess the relative importance of hierarchy and disentanglement, we additionally conduct an ablation study comparing: 1) our full model (learning both hierarchy and disentanglement on the HiVG dataset), 2) a model variant without hierarchical learning (i.e. hierachical loss), and 3) a model variant without disentanglement learning (i.e. Tride module and Tride loss). The results are summarized below.
> - To address the reproducibility concern, we confirm that we will open-source our code and trained model weights upon paper acceptance.
>
> | Model | D3 (Full)|OmniLablel|
> |------------------|-------|------|
> | Full model           | 30.7  | 26.9  |
> | w/o hierarchy        | 28.8  | 25.8  |
> | w/o disentanglement  | 28.6  | 26.2  |
>
> - As shown in the Table, disentanglement and hierarchy act as **complementary modules**, and each contributes to performance improvement. With disentanglement alone, learnable vectors separate text embeddings and increase angular distance between confusing captions (e.g., black segway vs. white segway), but the model does not learn positive and negative distinctions with the objects (e.g., segway)
> - On the other hand, without disentangling the shared semantic anchor (e.g., segway), the embedding space lacks stability and becomes difficult to structure coherently. In contrast, when hierarchy learning operates on disentangled embeddings, the shared semantic concept *Segway* is preserved as a reliable anchor, enabling more effective separation of fine-grained attribute differences.
>
> > **W4:** Minor Presentation Issues：
> > Some equations (e.g., Eq. 3, 6) are dense and could use more intuition.
> > Figures could label components more clearly (especially in `Fig. 2`).
> - We added intuitive explanations below Eq. 3 and Eq. 6 and improved clarity in `Fig. 2` by refining the component labels. We also redrew `Fig. 4` for better readability. These revisions are included in the updated manuscript.

---

> > ### Author Response · Authors · 2025-11-22
> > **Official Comment by Authors**
> >
> > > **Q1:** The final text embedding E=pooling(O+A+R). What pooling function is used (mean, [CLS] token, attention)? Have you compared different pooling methods to see how they affect compositional sensitivity?
> >
> > - Alignment with visual features is grounded through the **mean-pooling** token follwoing the same mean pooling strategy in GLEE. We have added this clarification to the revised manuscript (Lines 263–-264).
> >
> > > **Q2:** How is the “reference point” r in Eq. 6 chosen or updated?
> >
> > - For negative instances, the reference point corresponds to the positive embedding from the previous tier and is updated during training. In sentence-level negative pairs (e.g., white segway vs. black segway), the objects differ but share a common semantic component (e.g., segway). In such cases, the reference point used to enforce a larger angular separation should not be an arbitrary text-embedding centroid; instead, it must maintain an angular direction distinct from the shared semantic component.
> > - For negative examples, this reference point serves as a learnable anchor and is updated accordingly. In contrast, for positives, angular alignment is performed relative to a fixed central anchor, which is not updated.
> >
> > > **Q3:** Why were structure-aware or compositional baselines like CLIP-Adapter, LogicZSL, or MDETR not included?
> >
> > - We cited CLIP-Adapter to reference multi-modal contrastive learning, not compositional learning, and we will clarify this distinction in the revised version. LogicZSL uses LLMs to generate descriptions of semantic concepts, which aligns with the purpose of our baseline DesCo. However, because LogicZSL is designed for classification whereas DesCo is specifically tailored for language-based object detection, we adopt DesCo as our baseline. Regarding MDETR, although it was included as a baseline in OmniLabel, its performance on OmniLabel is substantially lower than that of our backbone GLEE, so we omit it from our comparisons. We will add it as a baseline in Table 1.
> > - We have revised the manuscript to clarify this point. Thank you for the helpful suggestion.
> >
> >
> > > **Q4:** Have you identified scenarios where the hierarchical structure fails — e.g., ambiguous relational clauses or overlapping attributes?
> >
> > - We provide additional analysis of the failure cases in `Sec. G.3` and `Fig. 23` of the Appendix, with a focus on long text queries.
> > - We have added a discussion of these limitations and included qualitative examples to clarify the current challenges of our model and potential avenues for improvement.

---

> ### Author Response · Authors · 2025-11-27
> **Official Comment by Authors**
>
> We kindly inform you that we have submitted our rebuttal and updated the paper.
>
> We appreciate your valuable feedback and suggestions. We revised the manuscript accordingly and believe that the concerns have been addressed. Please let us know if any additional clarification would be helpful.

---

### Official Review · Reviewer_19CP · 2025-10-31

**Soundness:** 3
**Presentation:** 3
**Contribution:** 3
**Rating:** 6
**Confidence:** 3

**Summary:**

This paper introduces TaSe (Talk in Pieces, See in Whole), a framework designed to enhance language-based object detection by explicitly modeling the hierarchical and compositional structure of language. It first introduces a disentanglement module to break down text representations into core components of objects, attributes, and relations ('Talk in Pieces'), using a novel module called TriDe. It then designs a hierarchical aggregation mechanism to learn structured, sentence-level embeddings from these disentangled parts ('See in Whole'). To facilitate this, the authors introduce HiVG, a new dataset generated by using an LLM to re-caption the Visual Genome dataset with an explicit three-tier hierarchical structure. Experiment results on challenging language-based object detection benchmarks demonstrate that linguistic compositionality is crucial for the task of language-driven object detection.

**Strengths:**

1. The TaSe framework is well-designed that competently integrates disentanglement via attention and hierarchical learning via a Euclidean-space objective.
2. It achieves clear performance improvements over a strong baseline on the OmniLabel and D³ benchmarks. The ablation studies provide useful insights into the function of each component of the proposed framework.
3. The writing is clear, and the presentation of figures and diagrams is good.

**Weaknesses:**

1. In the HiVG dataset generation process, does the diversity of the generated negative captions influence the final performance? For example, how would the results vary if different large language models (LLMs) were used for re-captioning?
2. The authors describe their approach as “lightweight”. it would be helpful to provide a quantitative analysis of the computational overhead.
3. Given the framework's strong reliance on the O-A-R structure, how does it perform if queries that violate or deviate from this structure (eg. comparative phrases or multi-step relations)?
4. The TriDe module uses learnable queries to disentangle objects, attributes, and relations. Can the authors provide some qualitative visualizations to demonstrate that the module indeed learns meaningful disentanglement?

**Questions:**

1. In Lines 103–104, the authors state that the proposed reverse abstraction process “effectively mitigates hallucination issues.” Could the authors provide quantitative evidence or more systematic analysis to support this claim?
2. Can TaSe generalize to other VLM models?
3. How does the baseline trained on HiVG perform with complex but non-hierarchically annotated language?

---

> ### Author Response · Authors · 2025-11-22
> **Official Comment by Authors**
>
> Thank you for your constructive review and valuable comment. We address each of your concerns below.
>
> > **W1:** In the HiVG dataset generation process, does the diversity of the generated negative captions influence the final performance? For example, how would the results vary if different large language models (LLMs) were used for re-captioning?
>
> - We agreed that exploring open-source and recently strong-performing models would be beneficial. In response, we generated an additional recaptioning dataset using QWEN-3 8B [G] and GPT-5-nano [H] following the same procedure, producing 10K samples. We present the training results based on this dataset in the table below.
>    - [G] Yang, An, et al. "Qwen3 technical report." arXiv preprint arXiv:2505.09388 (2025).
>    - [H] OpenAI. 2025. GPT-5 System Card. August 13. Available at: https://cdn.openai.com/gpt-5-system-card.pdf
>  - Time constraints limited the number of diverse settings, but GPT-based models showed larger performance gains in our evaluation. We expect that combining these captions could demonstrate more meaningful improvements. We will add these results in `Fig. 6` of the revised manuscript.
>
> | Model     | D3 (Full)|
> |-----------|----------|
> | Llama3-7B     | 30.7     |
> | Qwen3-8B [G]  | 29.2     |
> | GPT5-nano [H] | **30.9** |
>
>
> > **W2:** The authors describe their approach as “lightweight”. it would be helpful to provide a quantitative analysis of the computational overhead.
> Given the framework's strong reliance on the O-A-R structure, how does it perform if queries that violate or deviate from this structure (eg. comparative phrases or multi-step relations)?
>
> - We use three additional learnable queries derived from O–A–R, each requiring an additional cross-attention operation. As shown in Tab. 6, this increases the computation by only 1.9 M operations, demonstrating an efficient design with minimal overhead.
> - We have added the latency analysis as requested. On average, 10 iterations are executed, requiring 62.2 ms per inference, and our module step shows an additional 1.57 ms (2.3%) overhead. We have been included in the Appendix **Line 917**.
> - Given the O–A–R framework, we evaluated zero-shot performance as a function of text length to assess whether queries that deviate from this structure degrade performance. Our captions have an average length of 8.75 words (standard deviation 1.34), and texts exceeding approximately 13 words tend to deviate from the structure (see Appendix `Sec. A.3`). As shown in the table below, our model maintains improved performance across both longer and shorter text lengths. We provide the detailed results in Fig. 5 of the revised manuscript.
>
> | Model | D3-S | D3-M | D3-L | D3-XL | D3-XXL | Omni-S | Omni-M | Omni-L | Omni-XL | Omni-XXL |
> |--------|------|------|------|-------|--------|--------|--------|--------|---------|----------|
> | **GLEE** | 28.8 | 28.7 | 28.8 | 26.8 | 28.6 | 27.3 | 14.1 | 11.0 | 9.5 | 8.6 |
> | **Ours** | **30.0** | **31.1** | **31.2** | **28.1** | **29.0** | **32.6** | **19.4** | **17.2** | **13.3** | **11.3** |
>
> > **W3:** The TriDe module uses learnable queries to disentangle objects, attributes, and relations. Can the authors provide some qualitative visualizations to demonstrate that the module indeed learns meaningful disentanglement?
>
> - Thank you for the valuable feedback. We conducted additional experiments to examine the effects of hierarchical learning and disentanglement on the embedding space. For hierarchical learning, as shown in `Fig. 20` of the Appendix, we initialized embeddings from random points and evaluated them using positive and negative pairs to analyze the behavior of our method. The effects of disentanglement are visualized in `Fig. 16`, illustrating how the separated components form a structured embedding space.

---

> > ### Author Response · Authors · 2025-11-22
> > **Official Comment by Authors**
> >
> > > **Q1:** In Lines 103–104, the authors state that the proposed reverse abstraction process “effectively mitigates hallucination issues.” Could the authors provide quantitative evidence or more systematic analysis to support this claim?
> >
> > - Hallucination in LLMs can affect detection performance, so we implemented a mitigation strategy. As shown in `Fig. 10`, we generated recaptions from Tier 3 to Tier 1 using a special delimiter (=>). When the delimiter was missing, we applied filtering and regenerated the captions. This process reduced hallucinations, and we conducted a human evaluation to validate its effectiveness. The results show that 98.8% of the generated sentences aligned with the intended meaning, with only **1.2%** containing hallucination.
> > - Additional details are provided in Appendix `Sec. D.2 and `Fig. 14.
> >
> >
> > > **Q2:** Can TaSe generalize to other VLM models?
> >
> > - VLM detectors typically adopt dual-branch encoders—one for visual inputs and one for textual inputs—followed by an object decoder with multimodal fusion. Since most state-of-the-art detectors (*e.g.,* GLEE) rely on BERT-based text encoders which will share the same limitations, and our method enhances the compositional capability of these encoders, the proposed approach can be broadly applicable and benefit a wide range of VLM detection models. We observe that LLM-based text encoders exhibit similar limitations, as illustrated in `Fig. 22` of the Appendix. We further believe that our hard-negative generation pipeline and fine-grained discrimination strategy will be broadly applicable to other VLMs using diverse text encoder architectures.
> >
> > > **Q3:** How does the baseline trained on HiVG perform with complex but non-hierarchically annotated language?
> >
> > - Simply increasing text length for LoRA tuning did not yield meaningful performance gains. In addition, we trained the model with more robust supervision by incorporating negative captions, following the DesCo strategy. When trained on the same 10K dataset, the GLEE+DesCo results in `Tab. 1` show an improvement from 27.6 to 28.3 AP, with an additional gain of +0.9 for positive cases. Although this augmentation makes the model more resilient to negative captions, it still struggles to clearly distinguish positive from negative instances in complex language queries.
> > - To address this limitation, we applied hierarchy learning, focusing on generating both positive and negative captions. Our approach emphasizes determining whether objects are positive or negative using additional attribute and relation information, and improving sentence-level compositional reasoning.

---

> > > ### Author Response · Authors · 2025-11-28
> > > **Official Comment by Authors**
> > >
> > > We kindly inform you that we have submitted our rebuttal and updated the paper.
> > >
> > > We sincerely appreciate your valuable feedback and suggestions. At your convenience, we would be grateful if you could review the revisions and share any additional comments during the discussion phase. Please let us know if any further clarification would be helpful.

---

> > > ### Comment · Reviewer_19CP · 2025-11-28
> > > **The authors have addressed my concerns**
> > >
> > > I appreciate the author’s detailed and comprehensive experimental rebuttal. The new experiments and quantitative analyses, particularly those examining dataset robustness and zero-shot performance on complex queries, demonstrate the effectiveness of their method and address my concerns. I therefore maintain my rating of 6 and recommend acceptance.

---

### Official Review · Reviewer_PuqX · 2025-11-03

**Soundness:** 3
**Presentation:** 2
**Contribution:** 2
**Rating:** 6
**Confidence:** 3

**Summary:**

This paper aims to tackle the problem that VLM-based OVD models fall short in understanding complex queries. The authors make an observation that the text embedding of VLM can not well distinguish different attributes / relations for the same object, and consider this phenomenon as the main reason that causes the insufficient ability. To tackle this problem, they first construct a new data pipeline that has structured descriptions with "object - attribute - relation" hierarchy. Under this hierarchy, each data point (the bbox) has multiple hard negative text names that have different attribute / relation.  They also propose a new method that first disentangles  the embedding into three components (object / attribute / relation) and then re-aggregates them into a new embedding. The corresponding training method has some novelty for accommodating the hierarchy, as well.

**Strengths:**

- The observation on the text embedding well illustrates the reason that OVD-VLMs fail to understand complex queries.

- The solution is comprehensive, covering from the data construction, the model architecture design, as well as the corresponding training objectives.

- The data construction part is reasonable, and the method seems novel.

- The ablation study is comprehensive.

**Weaknesses:**

- The presentation and writing needs improvement. Some parts are difficult to follow.

- According to the example given in Section 3.1, the most import aspect of HIVG is to generate HARD negative text discription of each box, i.e., the description sharing the same object but have different attribute (Tier 2) or realtion (Tier 3). It would be more clear for the authors to describe this principle into the introduction.

- In Eq. 1,
    - 1) What is the text embedding X in Line 1?
    - 2) In Line 2, Which one is the query / value for the cross attention, respectively?
    - 3) It seems that TriDe not only disentangles the original embeddings, but also re-aggregates the derived O, A, R into the output E (Line 3). What is the motivation of this design?

- The improvement on D^3 dataset is very small, e.g., +1.0 FULL AP over the GLEE baseline.

- The employed datasets are insufficient. The authors have not includes a popolar dataset RefCoCo which also has long and fine-grained descriptions, and is popular for visual grounding evaluation. Is there any specific reason?

**Questions:**

- In the introduction, the authors reveal an observation that descriptions with partially-shared and partiall-different words have over-close embedding distances. However, recent SoTA VLMs are already trained with fine-grained discriptions. I am interested if the authors have deeper analysis on the reason why this phenomenon still happens.

- How does your method perform on general OVD dataset (e.g., LVIS). You may use the Out-of-Domain setting for easy comparison,i.e., you train a baseline method on your dataset and compare this baseline against your method.

- Will the proposed hierarchical and disengtangled representation learning be good for the VLM training (besides the OVD as in the paper)?

---

> ### Author Response · Authors · 2025-11-22
> **Official Comment by Authors**
>
> We sincerely appreciate the reviewer’s encouraging comments and thoughtful questions. We address each concern below.
>
> > **W1:**  The presentation and writing needs improvement. Some parts are difficult to follow.
> - We thank the reviewer for pointing this out. We will add further clarification and make the necessary revisions to improve readability.
>
> > **W2:** According to the example given in Section 3.1, the most import aspect of HIVG is to generate HARD negative text discription of each box, i.e., the description sharing the same object but have different attribute (Tier 2) or realtion (Tier 3). It would be more clear for the authors to describe this principle into the introduction.
> - Hard negative examples are indeed an important component in our setting, as they help distinguish fine-grained attribute or relation variations that share the same object identity. That said, the core goal of our approach is to learn sentence-level hierarchy through both positive and negative caption generation, enabling structured reasoning beyond hard negatives alone. Our intention is to show that a VLM text encoder—primarily aligned at the word level—can improve zero-shot grounding performance when trained to model hierarchical roles of objects, attributes, and relations.
> - To clarify this motivation and emphasize the significance of hierarchical learning, we have revised the introduction. We added clarification in the **Lines 98–102** regarding the definition of HiVG data and the motivation for constructing the hierarchy structure it provides.
>
> > **W3-1,2:** In Eq. 1, What is the text embedding X in Line 1? In Line 2, Which one is the query / value for the cross attention, respectively?
> - The text embedding X is obtained from the CLIP text encoder. We have updated the manuscript to clarify these points. Specifically, we added an explanation of the text embedding **Line 234 -- 235** and clarified the query and value roles in the cross-attention module in **Line 258 -- 260**.
>
>
> > **W3-3:** It seems that TriDe not only disentangles the original embeddings, but also re-aggregates the derived O, A, R into the output E (Line 3). What is the motivation of this design? Need clearer explanations and motivation for the components and design choices in Equation 1.
> - When processing complex language queries, existing VLMs often behave like bag-of-words models and fail to distinguish objects from their attributes and relations [A]. Therefore, we first disentangle these components. Since attribute and relation information is necessary to differentiate objects within the same category—as illustrated in `Figs. 1` and 6—we subsequently aggregate the disentangled components.
>    - [A] Yuksekgonul, Mert, et al. "When and why vision-language models behave like bags-of-words, and what to do about it?." arXiv preprint arXiv:2210.01936 (2022).
> - In addition, we observed that directly increasing the angular distance for negative captions caused shared textual content to push the embeddings apart, leading to a sharp increase in the overall loss. To address this issue, we disentangle and aggregate components to enable dynamic learning of angular information within the embeddings. We added an explanation clarifying the motivation behind this design in the revised manuscript **(Line 262 -- 264)**.
>
> > **W4:** The improvement on D³ dataset is very small, e.g., +1.0 FULL AP over the GLEE baseline. Limited performance gains on the D³ dataset.
> - We would like to respectfully clarify that the improvements on the D³ dataset are **not** marginal. As shown in the main metric (noted as Full) in Table 1, our TaSe framework yields a substantial performance gain of +3.1 AP, improving the GLEE-Lite baseline from 27.6 to 30.7. In comparison, the DesCo baseline achieve only a +0.7 AP gain.

---

> > ### Comment · Reviewer_PuqX · 2025-11-27
> > **The authors have addressed most of my previous concerns**
> >
> > The authors have addressed most of my previous concerns. They have supplemented the manuscript with sufficient experimental results​ that comprehensively validate the effectiveness of their method. Furthermore, the additional discussion on the data structure provides valuable insight​ for contrastive learning. I am pleased to recommend it for acceptance.

---

> > > ### Author Response · Authors · 2025-11-27
> > > **Thank you**
> > >
> > > We are glad to hear that your concerns have been addressed! We sincerely thank you for your insightful feedback. We would be grateful if you might consider revisiting your evaluation.

---

> ### Author Response · Authors · 2025-11-22
> **Official Comment by Authors**
>
> > **W5:** The employed datasets are insufficient. The authors have not includes a popolar dataset RefCoCo which also has long and fine-grained descriptions, and is popular for visual grounding evaluation. Is there any specific reason?
>
> - We evaluate our method under the language-based **zero-shot detection setting** following DesCo. In the original DesCo paper, evaluations were conducted on LVIS and OmniLabel because its backbone, GLIP, was **not** pre-trained on LVIS. In contrast, our backbone GLEE is pre-trained on diverse detection datasets, including RefCOCO and LVIS. Therefore, we evaluate our model on OmniLabel as well as an additional zero-shot benchmark, D3.
> - In response, we conducted additional zero-shot experiments on gRefCOCO [B] and OdinW [C], which were not included in GLEE’s training data. We observe consistent performance gains.
> - The results have been added to `Tabs. 7 and 8` of the Appendix in the revised manuscript. The score corresponds to Pr@0.5, defined as the ratio of samples with F1=1 under IoU≥0.5, where FN and FP indicate unmatched ground-truth and predicted boxes, respectively.
>    - [B] He, Shuting, et al. "Grec: Generalized referring expression comprehension." arXiv preprint arXiv:2308.16182 (2023).
>    - [C] Li, Liunian Harold, et al. "Grounded language-image pre-training." Proceedings of the IEEE/CVF conference on computer vision and pattern recognition. 2022.
>
> | Model | gRefCOCO-val | gRefCOCO-testA | gRefCOCO-testB |
> |-------|--------------|----------------|----------------|
> | GLEE  | 60.6         | 59.6           | 51.2           |
> | TaSe  | **61.2**     | **63.2**       | **56.3**       |
>
>
> > **Q1:** In the introduction, the authors reveal an observation that descriptions with partially-shared and partially-different words have over-close embedding distances. However, recent SoTA VLMs are already trained with fine-grained descriptions. I am interested if the authors have deeper analysis on the reason why this phenomenon still happens.
> - Thank you for your insightful comment. We hypothesize that this observation aligns with findings in [A], which show that VLMs tend to behave like bags-of-words. This behavior arises largely from the absence of hard negatives, allowing models to rely on shortcuts. Similarly, in common pre-training detection datasets, fine-grained hard negatives at the attribute- or relation-level are scarce. Furthermore, RefCOCO’s training data contain only a single target object per expression and lack negative expressions. For example, given the query “a man with a segway” but an image containing only a single person, the model can simply match the phrase “a man” to that region without needing to consider the “segway” part of the expression. In contrast, OmniLabel and D3 present more challenging conditions, including multi-target descriptions and negative expressions where no referred object exists in the image.
> - We conducted additional experiments using text embeddings from other SOTA VLMs, SigLIP [D] and E5-V [E]. We added the corresponding discussion to Appendix **Section H** and visualization results in `Fig. 22`. Neither model provides sentence-level embeddings well-suited for object detection, as they tend to embed different objects with similar attributes and relations closely together.
>    - [A] Yuksekgonul, Mert, et al. "When and why vision-language models behave like bags-of-words, and what to do about it?." arXiv preprint arXiv:2210.01936 (2022).
>    - [D] Zhai, Xiaohua, et al. "Sigmoid loss for language image pre-training." Proceedings of the IEEE/CVF international conference on computer vision (2023).
>    - [E] Jiang, Ting, et al. "E5-v: Universal embeddings with multimodal large language models." arXiv preprint arXiv:2407.12580 (2024).
>
> > **Q2:** How does your method perform on general OVD dataset (e.g., LVIS). You may use the Out-of-Domain setting for easy comparison,i.e., you train a baseline method on your dataset and compare this baseline against your method.
>
> - As GLEE is pre-trained LVIS, we conducted zero-shot evaluation on the out-of-domain ODinW dataset [C] to evaluate performance in the OVD domain. ODinW is a real-world child dataset introduced by GLIP, constructed for zero-shot settings involving novel object categories and external knowledge descriptions. Our results show an improvement in zero-shot performance, with AP increasing from 44.4 to 49.3. We add these results to the `Tab. 8` of the Appendix.
>     - [C] Li, Liunian Harold, et al. "Grounded language-image pre-training." Proceedings of the IEEE/CVF conference on computer vision and pattern recognition. 2022.
>
> | Model | ODinW (Avg) |
> |-------|-------------|
> | GLEE  | 44.4        |
> | TaSe  | **49.3**    |

---

> > ### Author Response · Authors · 2025-11-22
> > **Official Comment by Authors**
> >
> > > **Q3:** Will the proposed hierarchical and disentangled representation learning be good for the VLM training (besides the OVD as in the paper)?
> >
> > - We believe our pipeline provides broader utility for VLMs beyond detection. In tasks where hard negatives are limited—as is common in complex reasoning benchmarks—hierarchical hard-negative generation strategies like HiVG can provide essential training signals. Moreover, for fine-grained understanding, separating key semantic components and aggregating them at the sentence level is highly effective for handling complex reasoning. We leave a deeper investigation of this aspect to future work.

---

### Official Review · Reviewer_z1ZV · 2025-11-04

**Soundness:** 3
**Presentation:** 3
**Contribution:** 3
**Rating:** 4
**Confidence:** 3

**Summary:**

Current VLMs struggle with complex language queries involving descriptive attributes and relational clauses. Their text encoders often behave like bags-of-words, failing to distinguish target objects from their attributes and relations, which leads to frequent false positives (e.g., confusing "segway with a man" for just "segway"). The paper proposes a framework designed to restructure linguistic representations according to hierarchical relations within sentences. The core principle is to disentangle text tokens into components and aggregate them into hierarchically structured sentence-level representations. The paper proposes:

HiVG Dataset: A hierarchical synthetic captioning dataset derived from Visual Genome. It structures captions into three tiers using a "reverse abstraction" process to mitigate hallucination:

A module that transforms text embeddings into three distinct subspaces, objects, attributes, and relations, guided by a disentanglement loss function.

A learning method that aggregates these disentangled components into structured embeddings. Unlike contrastive learning, it models full sentence hierarchy using dynamic references to enforce separation between negative pairs while preserving intrinsic embedding structure.

**Strengths:**

The paper directly tackles the "bag-of-words" behavior of current VLMs, which often fail to distinguish target objects from descriptive attributes and relations in complex queries.


The proposed TaSe framework demonstrates significant performance improvements over strong baselines like GLEE. It achieved a 24% performance improvement on the OmniLabel benchmark and increased AP scores by +3.1 on D³ and +5.2 on OmniLabel.

It introduces a framework that disentangles text into three core components (objects, attributes, relations) and aggregates them hierarchically, strengthening the inductive bias for linguistic structures.


The paper proposes the HiVG dataset.

**Weaknesses:**

1: The proposed HiVG dataset and TriDe module strictly enforce a three-tier hierarchy: Object, Attribute and Relation. While this addresses simple compound queries, natural language often involves deeper, recursive nesting that exceeds three tiers (e.g., "the man holding the cup that is on the table"). The current framework may struggle to represent these deeper recursive structures efficiently because it is designed purely around these three predefined components.

2. Ambiguity in "Attribute" vs. "Relation" Definitions. The distinction between Tier 2 (Attribute) and Tier 3 (Relation) appears conflated in the examples provided. For instance, "middle woman" is Tier 2, while "middle woman with dark hair" is classified as Tier 3 (Relation). Linguistically, "dark hair" is often considered an attribute of the entity, similar to "middle." By contrast, other Tier 3 examples involve distinct objects, such as "segway with a man". Treating intrinsic attributes (like hair color) and extrinsic relations (like accompanying objects) identically in Tier 3 might limit the model's ability to disentangle complex scenes where both distinct object interactions and fine-grained attributes are present simultaneously.

3. The fine-tuning relies on a relatively small synthetic dataset. The authors explicitly state they "use only HiVG dataset for training, which contains 10 K hierarchy captions". While efficient (training only 2.93% of parameters) , basing generic open-vocabulary improvements on such a small, specifically structured dataset risks overfitting to the specific syntactic patterns generated by Llama 3 during the re-captioning process. The claimed robustness to general complex queries might be limited to the specific domain covered by these 10K Visual Genome-derived captions.


4. The qualitative results presented focus exclusively on successes where TaSe outperforms the GLEE baseline by reducing false positives. There is no analysis of where the strict hierarchical enforcement might cause errors—for example, in idiomatic expressions that do not neatly decompose into object-attribute-relation, or where disentanglement might over-segment a coherent concept.

**Questions:**

See weaknesses.

---

> ### Author Response · Authors · 2025-11-22
> **Official Comment by Authors**
>
> Thank you for your valuable time to thoroughly review our work. We address each of your concerns below.
>
> > **W1:** The proposed HiVG dataset and TriDe module strictly enforce a three-tier hierarchy: Object, Attribute, and Relation. While this addresses simple compound queries, natural language often involves deeper, recursive nesting that exceeds three tiers (e.g., "the man holding the cup that is on the table"). The current framework may struggle to represent these deeper recursive structures efficiently because it is designed purely around these three predefined components.
>
> - Thank you for highlighting this important point. The objective of this work is to detect visual objects grounded in text. Rather than modeling the full syntactic structure of complex sentences, we focus on the elements that directly contribute to detection performance. In this context, the three components—**object, attribute, and relation**—provide an effective basis for grounding textual descriptions of visual entities. Hierarchy learning with these three components helps the model learn to separate positive and negative instances with finer semantic distinctions.
> - We acknowledge that this may not be the optimal way to perform the decomposition. Natural language allows much finer linguistic categorization. That said, our goal is to adopt a decomposition that enables models to parse descriptive sentences into visually grounded components that are effective for detection. Empirically, this scheme has been shown to improve object grounding performance, as demonstrated in our experiments.
> - We believe that exploring finer-grained semantic units for sentence-level hierarchy could be a meaningful direction for future work. We appreciate your insightful suggestion, which provides valuable guidance for future research.
> - We provide a detailed explanation of our evaluation across a wide range of text lengths—which may involve more complex linguistic structures—in the following response.
>
> > **W2:** Ambiguity in "Attribute" vs. "Relation" Definitions. The distinction between Tier 2 (Attribute) and Tier 3 (Relation) appears conflated in the examples provided. For instance, "middle woman" is Tier 2, while "middle woman with dark hair" is classified as Tier 3 (Relation). Linguistically, "dark hair" is often considered an attribute of the entity, similar to "middle." By contrast, other Tier 3 examples involve distinct objects, such as "segway with a man". Treating intrinsic attributes (like hair color) and extrinsic relations (like accompanying objects) identically in Tier 3 might limit the model's ability to disentangle complex scenes where both distinct object interactions and fine-grained attributes are present simultaneously.
>
> - In this work, we define components from the perspective of object detection as follows:
>    - Attribute: intrinsic properties of a single object (e.g., color, size, material).
>    - Relation: interactions or spatial/semantic connections between two or more objects.
>  - As you pointed out, certain expressions fall into a gray area from a linguistic perspective. For example, in “middle woman with dark hair”, “dark hair” can be interpreted as an attribute of “a man.” However, because hair appears as a visually separable part attached to the primary object, we consider such cases more suitable to categorize as relations when defining visual grounding and training detectors.
> - Accordingly, we explicitly disentangle descriptions into **object**, **attribute**, and **relation** layers so that the detector can attend to visual evidence corresponding to each layer. Although this three-layer structure does not capture every linguistic phenomenon, most visually groundable cues fall into these categories. Experimental results in `Tab. 1` show consistent improvements in object grounding when learning with this decomposition, **with stronger gains for longer and more complex sentences**.
> - In additional experiments **(see the table below or `Fig. 5`)**, comparing performance **across different sentence-length groups**, we observe that this improvement is consistent across sentence lengths and complexities. These results indicate that decomposing text into three components is an effective strategy for supporting accurate object grounding.
> - We have added clarification of this definition and the corresponding rationale in the revised manuscript.
>
> | Model | D3 (Small) | D3 (Medium) | D3 (Large) | D3 (Very Large) | Omnilabel (Small) | Omnilabel (Medium) | Omnilabel (Large) |
> |-------|-------|-------|-------|-------|-------|-------|-------|
> | GLEE  | 30.0   | 27.6    | 26.9   | 17.2   | 28.4   | 13.8  | 10.3
> | TaSe  | **31.8 (+1.8)**  | **31.2 (+3.6)**   | **30.3 (+3.4)**  | **19.8 (+2.6)** |  **33.1 (+4.7)**  | **19.3 (+5.5)** | **14.8 (+4.5)**  |

---

> > ### Author Response · Authors · 2025-11-22
> > **Official Comment by Authors**
> >
> > > **W3:** The fine-tuning relies on a relatively small synthetic dataset. The authors explicitly state they use only HiVG dataset for training, which contains 10 K hierarchy captions". While efficient (training only 2.93% of parameters) , basing generic open-vocabulary improvements on such a small, specifically structured dataset risks overfitting to the specific syntactic patterns generated by Llama 3 during the re-captioning process. The claimed robustness to general complex queries might be limited to the specific domain covered by these 10K Visual Genome-derived captions.
> >
> > - Thank you for the insightful question. We employed pre-trained models that were trained on diverse datasets, including RefCOCO, VisualGenome, LVIS, TAO, and OVIS. Although these datasets cover a broad range of linguistic and visual contexts, this powerful model still struggles to fully understand the semantic structure of complex sentences.
> > - Please note that our evaluation benchmarks, OmniLabel and D3, are designed for zero-shot evaluation and do not provide accompanying training data. Our approach shows that learning sentence hierarchy via fine-tuning on only 10 K carefully selected synthetic data can enhance **zero-shot** generalization performance.
> > - We have additionally validated our approach on zero-shot settings using gRefCOCO and OdinW datasets, and the results have been added to Appendix `Tab. 7` (grefCOCO) and `Tab. 8` (OdinW).
> >
> > > **W4:** The qualitative results presented focus exclusively on successes where TaSe outperforms the GLEE baseline by reducing false positives. There is no analysis of where the strict hierarchical enforcement might cause errors—for example, in idiomatic expressions that do not neatly decompose into object-attribute-relation, or where disentanglement might over-segment a coherent concept.
> >
> > - We provide additional analysis of the failure cases in `Sec. G.3` and `Fig. 23` of the Appendix, with a focus on long text queries.
> > - We have added a discussion of these limitations and included qualitative examples to clarify the current challenges of our model and potential avenues for improvement.

---

> ### Author Response · Authors · 2025-11-27
> **Official Comment by Authors**
>
> We kindly inform you that we have submitted our rebuttal and updated the paper.
>
> We sincerely thank you for your thoughtful feedback and suggestions. As the discussion phase approaches its end, we would appreciate it if you could let us know whether the updated manuscript and additional experimental results adequately address your concerns. If there are any further questions or suggestions, we would be glad to provide additional clarification.

---

### Author Response · Authors · 2025-11-23
**General Response**

Dear Reviewers, Area Chairs, Program Chairs, and Senior Area Chairs,

Thank you for taking the time to thoroughly review our paper and valuable suggestions. The primary updates are summarized below, with all modifications highlighted in blue in the revised version:
- **Clarified the motivation and refined presentation:** We made several adjustments throughout the manuscript to enhance readability and technical rigor, including improvements to the conceptual framing and overall presentation.
- **Additional analyses of the HiVG dataset:** We included distribution statistics and human evaluation results assessing caption quality and hallucination rate (`Appendix A`).
- **Extended zero-shot evaluations:** We added results on gRefCOCO and ODinW, demonstrating robustness on out-of-domain benchmarks in the zero-shot setting (`Appendix D.1`).
- **Expanded experiments with LLM-generated re-caption datasets:** We generated captions using two additional LLMs and compared models trained with three re-caption datasets, including dataset statistics (`Fig. 6` and `Fig. 12`).
- **Added analysis of model behavior:** We analyzed VLM embedding behavior (`Appendix E`), evaluated performance across sentence-length groups (`Fig 5`), and analyzed failure cases (`Appendix F.3`).
- **Expanded ablation and efficiency studies:** We quantified the contributions of hierarchical learning and disentanglement (`Tab. 2`), and reported latency and computational cost (`Appendix B.1`).

Below, we provide detailed, point-by-point responses to each reviewer’s comments. Thank you again for your thoughtful feedback. We hope that the revisions sufficiently address the concerns raised.

---

### Author Response · Authors · 2025-12-01
**Discussion Summary by Authors**

Dear Area Chair,

We would like to briefly summarize the discussion that took place during the rebuttal period. Reviewers raised concerns regarding:
- Hallucination issues in our hierarchically generated caption dataset and the effectiveness of disentangled representation learning;
- Whether recent VLMs exhibit bag-of-words-like behavior, revealing limitations in current disentanglement approaches;
- The value of additional evaluations on open-vocabulary detection benchmarks.

We addressed these concerns thoroughly in our response. Reviewers **PuqX** and **19CP** commented that their initial concerns were reasonably clarified and recommended the paper for acceptance. Although we did not receive follow-up comments from **z1ZV** and **MTQ7**, their concerns aligned with previously discussed points, and we believe they were addressed in the same context.

To support our response, we conducted additional experiments, including:
- Expanded evaluation of the HiVG dataset and LLM-based recaption datasets (Qwen3, GPT);
- Embedding analysis of recent VLMs (Qwen-VL, E5-V) and failure-case studies;
- Evaluation of OOD generalization on OmniLabel and D³, with additional results on ODinW and gRefCOCO.

All corresponding improvements have been added into the revised manuscript.
Thank you very much for taking the time to review our work.

---

### Meta-Review · Area_Chair_2Jpr · 2026-01-06

**Summary:**

Current VLMs still struggle with complex text queries that include complex descriptive attributes. To address this, the paper proposes a framework that introduces HiVG, a hierarchical synthetic caption dataset built from Visual Genome, and learns disentangled linguistic representations on object, attribute, and relation, which are then hierarchically aggregated into structured sentence-level embeddings for improved grounding.

Reviewers generally agree that the paper identifies a limitation of current VLM-based grounding/detection, with clear gains on challenging benchmarks.
Beyond requests for more benchmarks to strengthen generalization claims, implementation details, finer clarifications of HiVG negative-caption construction, and additional failure-case analysis, **the main concerns are:**

1. **HiVG dataset analysis and generalization.** Reviewers ask for more quantitative analysis of HiVG (diversity/quality/balance), more systematic evidence that the reverse-abstraction pipeline mitigates hallucinations, and they question whether training on a small, synthetic LLM-generated dataset may limit generalization or risk overfitting to specific captioning patterns (MTQ7, 19CP, z1ZV).

2. **Limitations of the proposed O–A–R structure.** Reviewers note that the object–attribute–relation decomposition may have overlaps/ambiguities (e.g., attribute vs. relation boundary) and may be insufficient for natural language that exhibits deeper recursive nesting or multi-step compositional structures, potentially limiting applicability to more complex queries (z1ZV, 19CP).

**Reviewer Concerns:**

The rebuttal improved clarity and completeness by clarifying key implementation details (e.g., pooling, query/value roles), providing quantitative evidence on HiVG caption quality and hallucination rate via human evaluation, expanding zero-shot evaluations to additional OOD benchmarks (e.g., gRefCOCO, ODinW), reporting ablations to separate the effects of hierarchy vs. disentanglement, and quantifying overhead with latency analysis.

However, some concerns are only partly addressed in the rebuttal:

1. Fixed O–A–R hierarchy (Reviewers z1ZV and 19CP): Sentence-length analyses provide indirect evidence, but the three-tier decomposition may still be limited for deeper recursive structures or queries that deviate from the O–A–R form.

2. Attribute vs. relation boundary (Reviewer z1ZV): The rebuttal clarifies detection-oriented definitions, but it remains unclear whether ambiguous cases (e.g., “dark hair”) could lead to mis-grouping that introduces errors or weakens disentanglement in complex scenes.

3. Sensitivity to small synthetic HiVG (Reviewer z1ZV): Additional OOD results are helpful, yet it is still unclear how dependent the gains are on the specific LLM re-captioning style or syntactic patterns in the 10K captions.

**Reviewer Scores:**

For reviewer PuqX (score 6): likely keep 6. The reviewer stated that most concerns were addressed and recommended acceptance after the discussion.

For reviewer 19CP (score 6), likely keep 6. The reviewer said, "I therefore maintain my rating of 6 and recommend acceptance."

For reviewer z1ZV (score 4), likely keep 4. While the rebuttal clarifies the attribute/relation definitions and adds extra zero-shot results, the main concern about the structure limitation and potential overfitting to the 10K LLM-generated captions is not fully resolved.

For reviewer MTQ7 (score 4), likely keep 4 or raise to 6. The rebuttal adds HiVG quality analysis, human evaluation, and stronger ablations, but concerns about the cumulative multi-module complexity remain.

---

### Decision · Program_Chairs · 2026-01-26

Reject